# Identification of a weight loss-associated causal eQTL in *MTIF3* and the effects of *MTIF3* deficiency on human adipocyte function

**Mi Huang[1], Daniel Coral[1], Hamidreza Ardalani[2], Peter Spegel[2], Alham Saadat[3], Melina Claussnitzer[3], Hindrik Mulder[4], Paul W Franks[1,5]\*†, Sebastian Kalamajski[1]\*†**

[1]Genetic and Molecular Epidemiology Unit, Department of Clinical Sciences, Clinical Research Centre, Lund University, Malmö, Sweden; [2]Department of Chemistry, Centre for Analysis and Synthesis, Lund University, Lund, Sweden; [3]Metabolism Program, Broad Institute of MIT and Harvard, Cambridge, United States; [4]Unit of Molecular Metabolism, Department of Clinical Sciences, Clinical Research Centre, Lund University, Malmö, Sweden; [5]Department of Nutrition, Harvard T.H. Chan School of Public Health, Boston, United States

**\*For correspondence:**
paul.franks@med.lu.se (PWF);
sebastian.kalamajski@med.lu.
se (SK)

†These authors contributed
equally to this work

**Competing interest:** The authors declare that no competing interests exist.

**Abstract** Genetic variation at the *MTIF3* (Mitochondrial Translational Initiation Factor 3) locus has been robustly associated with obesity in humans, but the functional basis behind this association is not known. Here, we applied luciferase reporter assay to map potential functional variants in the haplotype block tagged by rs1885988 and used CRISPR-Cas9 to edit the potential functional variants to confirm the regulatory effects on *MTIF3* expression. We further conducted functional studies on MTIF3-deficient differentiated human white adipocyte cell line (hWAs-iCas9), generated through inducible expression of CRISPR-Cas9 combined with delivery of synthetic *MTIF3*-targeting guide RNA. We demonstrate that rs67785913-centered DNA fragment (in LD with rs1885988, $r^2$ > 0.8) enhances transcription in a luciferase reporter assay, and CRISPR-Cas9-edited rs67785913 CTCT cells show significantly higher *MTIF3* expression than rs67785913 CT cells. Perturbed *MTIF3* expression led to reduced mitochondrial respiration and endogenous fatty acid oxidation, as well as altered expression of mitochondrial DNA-encoded genes and proteins, and disturbed mitochondrial OXPHOS complex assembly. Furthermore, after glucose restriction, the *MTIF3* knockout cells retained more triglycerides than control cells. This study demonstrates an adipocyte function-specific role of *MTIF3*, which originates in the maintenance of mitochondrial function, providing potential explanations for why *MTIF3* genetic variation at rs67785913 is associated with body corpulence and response to weight loss interventions.

## Editor's evaluation

In this study, Huang et al. perform detailed functional genomics assays in cultured adipocytes to provide mechanistic insight underlying an important obesity GWAS locus. These studies not only demonstrate allele-specific effects of MITF3 as a potential causal gene for variations in rs67785913 and rs1885988 alleles, but further provide a foundational framework from bridging GWAS associations to actionable pathways. The study strengths include genetic manipulation followed by detailed biochemical characterization to mimic and test the impacts of association, where future studies potentially involving in vivo characterizations could further inform the metabolic consequences of these observations.

## Introduction

Over 650 million people are obese and often suffer from metabolic abnormalities, including dyslipidemia, type 2 diabetes, and hypertension (*Adams et al., 2006*; *May et al., 2020*). It is widely believed that obesity results from an interplay between genetic and environmental factors (*Thomas, 2010*), but the biological mechanisms behind these interactions are poorly understood.

Genetic variation (rs12016871) at *MTIF3* (encoding the Mitochondrial Translation Initiation Factor 3 protein [*Kuzmenko et al., 2014*]) has been robustly associated with body mass index (BMI) in humans (*Locke et al., 2015*). Several subsequent studies have linked *MTIF3* genetic variation with the response to weight loss interventions, including diet, exercise, and bariatric surgery (*Papandonatos et al., 2015*; *Rasmussen-Torvik et al., 2015*), and with weight-related effects of habitual diet (*Nettleton et al., 2015*). For example, analyses in two of the world's largest randomized controlled weight loss trials (Diabetes Prevention Program [DPP] and Look AHEAD) found that homozygous minor allele carriers (rs1885988) were slightly more prone to weight gain in the control arm, yet achieved significantly greater weight loss at 12-month post-randomization and retained lost weight longer (18–36 months) than major allele carriers (*Papandonatos et al., 2015*). Elsewhere, the same locus has been associated with greater and more sustained weight loss following bariatric surgery (*Rasmussen-Torvik et al., 2015*).

Mtif3 loss in the mouse results in cardiomyopathy owing to impaired translation initiation from mitochondrial mRNAs and uncoordinated assembly of OXPHOS complexes in heart and skeletal muscle (*Rudler et al., 2019*). In the human hepatocyte-like HepG2 cell line, MTIF3 loss decreases the translation of the mitochondrial-encoded ATP synthase membrane subunit 6 (*ATP6*) mRNA without affecting cellular proliferation (*Chicherin et al., 2020*). No human genomic mutations leading to total MTIF3 deficiency have been reported, but the studies outlined above suggest that *MTIF3* may influence obesity predisposition and weight loss potential by modulating mitochondrial function; thus, *MTIF3* may play a key role in adipose tissue metabolic homeostasis, as adipocyte mitochondria not only provide ATP, but also impact adipocyte-specific biological processes such as adipogenesis, lipid metabolism, thermogenesis, and regulation of whole-body energy homeostasis (*Gregoire et al., 1998*; *Boudina and Graham, 2014*).

In this study, we aimed to experimentally dissect the molecular mechanisms that could underlie the correlation between *MTIF3* genetic variation and weight loss intervention outcomes. We hypothesized that among the common genetic variants in *MTIF3*, one (or more) is causal for altered *MTIF3* expression. Secondly, we hypothesized that MTIF3 content in human white adipocytes influences adipocyte-specific, obesity-related traits under basal and perturbed metabolic conditions. For the latter, we used glucose restriction to mimic the effects of in vivo lifestyle interventions focused on energy restriction and expenditure.

## Results

### rs67785913 is a regulatory variant for *MTIF3* expression

The *MTIF3* rs1885988 C allele is associated with enhanced weight loss and weight retention following lifestyle intervention trials in DPP and Look AHEAD cohorts (*Papandonatos et al., 2015*). In the GTEx database, the rs1885988 associates with an eQTL in subcutaneous fat (*Figure 1A*), with C allele carriers having significantly higher *MTIF3* expression (normalized effect size: 0.15, p = 0.0000032).

To experimentally validate and fine map the potential causal DNA variation in the haplotype block tagged by rs1885988, we looked up all tightly linked ($r^2 > 0.8$) single-nucleotide polymorphisms (SNPs) in HaploReg database v4.1 (*Ward and Kellis, 2012*). We then PCR-amplified and cloned 12 DNA fragments from that haploblock, altogether comprising the linked SNP loci, into luciferase reporter plasmids. As shown in *Figure 1D*, by comparing the luciferase signals with minimal promoter (minP) construct, only one DNA fragment (F11), encompassing the rs67785913 locus, could enhance luciferase transcription. Coincidentally, the rs67785913 also shows an eQTL effect on *MTIF3* expression in subcutaneous adipose tissue in GTEx database, with the major CT allele associated with lower expression than the minor CTCT allele (normalized effect size: −0.16, p = $3.0 \times 10^{-8}$) (*Figure 1B*). To demonstrate an allele-specific regulatory effect on *MTIF3* expression, we then used CRISPR-Cas9 to substitute the major CT for the minor CTCT allele at the rs67785913 locus in the pre-adipocyte hWAs cell line. Due to rather low CRISPR editing efficiency of that locus, we needed to genotype over

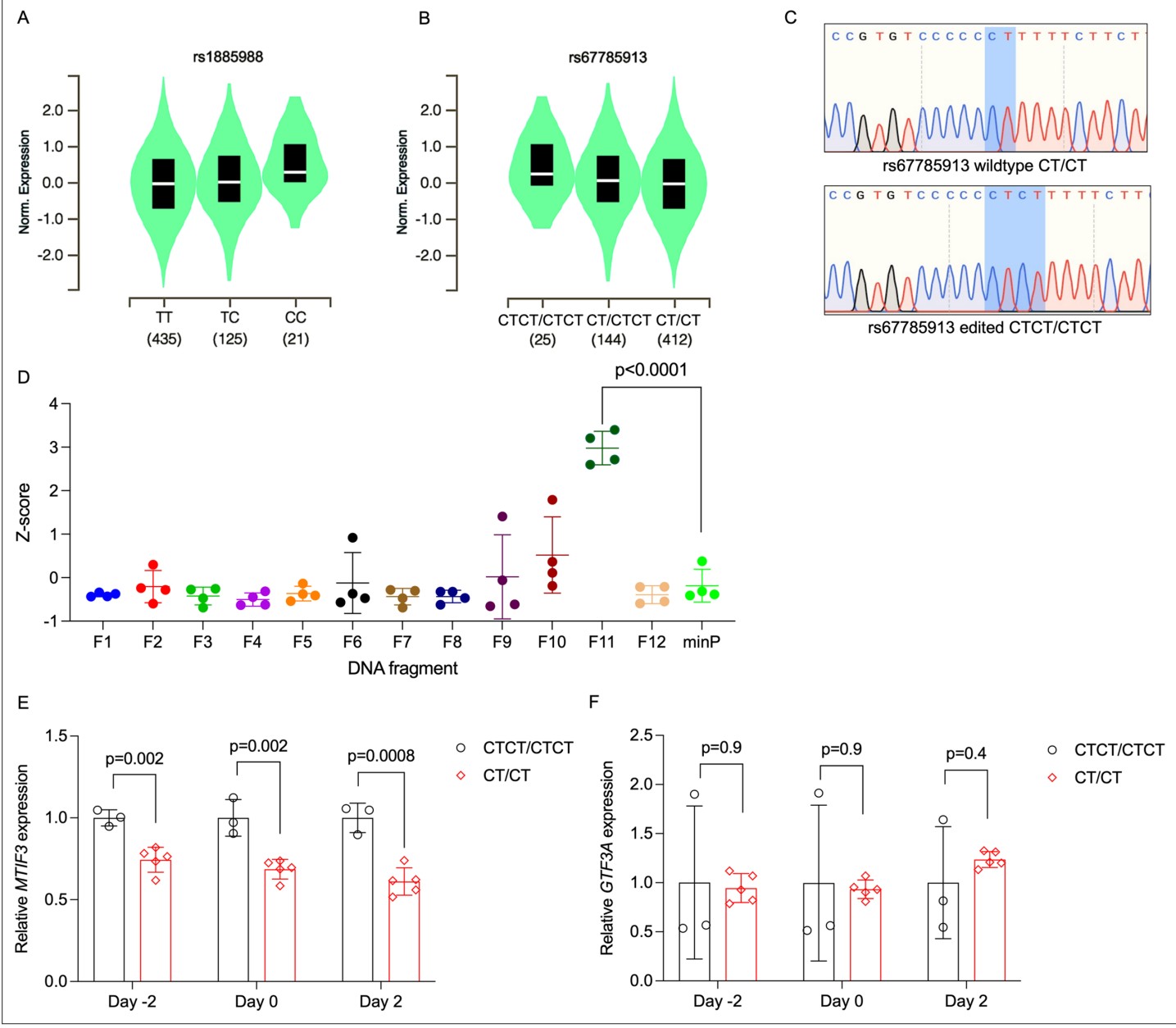

**Figure 1.** Identification of rs67785913 as a causal cis-eQTL for *MTIF3*. (**A**) Violin plot of *MTIF3* expression in subcutaneous adipose tissue for rs1885988 from Genotype-Tissue Expression (GTEx) Project eQTL. (**B**) Same as in (**A**), but for rs67785913. (**C**) Representative Sanger sequencing traces of rs67785913 CTCT/CTCT and CT/CT clones obtained after CRISPR/Cas9-mediated allele editing and single-cell cloning. (**D**) Normalized *Z*-score plot of luciferase reporter assays using vectors carrying different DNA fragments of the *MTIF3* gene cloned into pGL4.23 luciferase reporter vector. Hypothesis testing was performed by comparing the transcriptional enhancer activity of each of the 12 vectors (F1–12) to the empty vector (minP). All data were plotted as mean ± standard deviation (SD), *n* = 4 independent experiments, p values are presented in each graph; ordinary one-way analysis of variance (ANOVA) was used for statistical analysis. (**E**) Relative *MTIF3* expression (mRNA) in rs67785913 allele-edited cells 2 days before, at, or 2 days post-differentiation induction (day −2, 0, and 2, respectively). *n* = 3 clonal populations for CTCT/CTCT genotype, *n* = 5 clonal populations for CT/CT genotype, error bars show SD. (**F**) as in (**E**), but for *GTF3A* (mRNA) expression. Two-tailed Student's *t*-test was used; p values are presented in each graph.

The online version of this article includes the following figure supplement(s) for figure 1:

**Figure supplement 1.** To test if rs67785913 affects adipogenic differentiation in hWAs cells, we differentiated the rs67785913 allele-edited cells (CTCT/CTCT vs. CT/CT) for 12 days.

700 single-cell clones to obtain five CT/CT and three CTCT/CTCT clones without random indels, as confirmed by Sanger sequencing (*Figure 1C*). We then examined *MTIF3* expression in these clones at pre- and post-adipogenic differentiation induction, and found rs67785913 CTCT/CTCT to confer higher *MTIF3* expression at all time points (*Figure 1E*), although without apparent change on adipogenic differentiation markers (*Figure 1—figure supplement 1*). As rs67785913 also correlates with an altered *GTF3A* expression in other tissues (e.g., muscle, lung), we also detected, but found no apparent difference in *GTF3A* expression in rs67785913-edited cells (*Figure 1F*).

## Generating inducible Cas9-expressing pre-adipocyte cell line (hWas-iCas9)

Next, we intended to use the rs67785913-edited cells in functional genomics experiments to examine the phenotypic consequences of the eQTL. To conduct meaningful studies of gene × environment

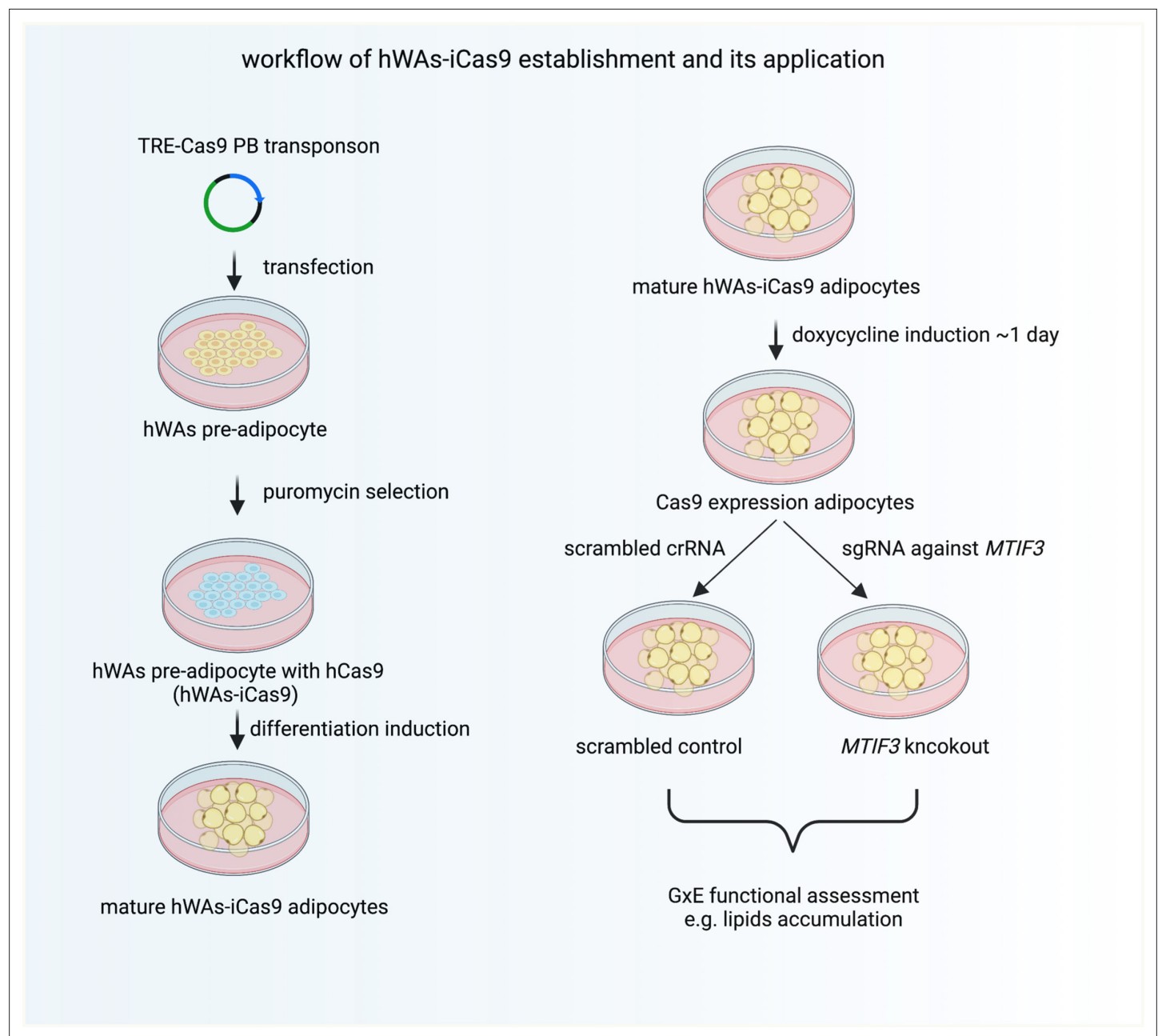

**Figure 2.** The workflow of establishing hWAs-iCas9 cell line and its application in studying MTIF3 and environment interactions in vitro.

interaction, it is desirable to use similarly differentiated cells with comparable baselines (e.g., similar triglyceride or mitochondrial content). Unfortunately, marginally different passage numbers between control and experimental groups can confound adipogenic differentiation. This problem can originate during single-cell cloning to create genetic knockouts/knockins, and became apparent with our rs67785913 allele-edited cells. While the mean values of adipogenic markers were similar in both rs67785913 genotypes (*Figure 1—figure supplement 1*), the variation between clones of the same genotype precluded the use of these cells in gene × environment studies. To circumvent this, we instead established an inducible Cas9-expressing pre-adipocyte cell line that allowed us to knockout *MTIF3* after completed adipogenic differentiation. As illustrated in *Figure 2*, we co-transfected hWAs with two plasmids: one encoding piggyBac transposase, and the other carrying piggyBac transposon-flanked doxycycline-inducible Cas9 and constitutively expressed puromycin resistance genes. In this setup, piggyBac transposase drives the integration of the piggyBac-transposon-flanked genes, and transgenic cells are then selected and expanded in puromycin-supplemented culture medium. We have thus obtained an hWAs cell line with doxycycline-inducible Cas9 expression and maintained adipogenic differentiation capacity (henceforth called hWAs-iCas9). The Cas9-expressing differentiated cells could then be transfected with relatively low molecular weight synthetic single guide RNAs (sgRNAs) that complex with intracellularly expressed Cas9 and target the gene exon of interest to generate random indels (in essence, gene knockouts). We used this method here to determine the functional role of *MTIF3* in adipocyte biology.

## Generation of *MTIF3* knockout in hWAs-iCas9 mature adipocytes

To investigate the role of *MTIF3* in human adipocyte development and energy metabolism we generated stable *MTIF3* knockouts in differentiated hWAs-iCas9 adipocytes. We designed Cas9-specific sgRNA to generate random indels in the exon expressed in all three *MTIF3* protein-encoding transcripts (*Figure 3A*) and obtained a >80% reduction in MTIF3 protein levels in every experiment, as assessed by western blotting (*Figure 3B–D*, *Figure 3—figure supplement 1*, and *Figure 4A, B*). To assess off-target effects of CRISPR-Cas9, we also performed T7EI assays on PCR-amplified top 5 predicted off-target sites and did not observe any detectable off-targeting (data are not shown).

## *MTIF3* knockout in mature adipocytes does not affect adipogenic marker or lipid content

Although the *MTIF3* knockout in hWAs-iCas9 cells was generated *after* the cells were differentiated, we wanted to ensure the genetic perturbation did not affect adipogenic markers or triglyceride content, as that could confound results from downstream functional studies. Incidentally, we observed that the quantities of the adipogenic markers, including ACC, FABP4, and FAS were comparable in control and *MTIF3* knockout cells (*Figure 3C, E*; see also *Figure 3—figure supplement 1*). Similarly, there were no apparent differences in Oil-red O or total triglyceride content (*Figure 3F, G*).

## *MTIF3* knockout disrupts mitochondrial DNA-encoding gene and protein expression, mitochondrial content, as well as mitochondrial OXPHOS assembly in hWAs-iCas9 adipocytes

MTIF3 is a mitochondrial translation initiation factor; thus, we examined the effects of MTIF3 ablation on differentiated hWAs adipocyte mitochondrial respiration chain. Assessed by western blotting, the *MTIF3* knockout cells had significantly decreased COX II (subunit of OXPHOS complex IV) and ND2 (subunit of OXPHOS complex I), trending decrease of CYTB (subunit of OXPHOS complex III), and unchanged ATP8 (subunit of OXPHOS complex V) content (*Figure 4A , B*). Moreover, using qPCR, we observed an altered expression of several mitochondrial DNA-encoding genes. Specifically, MTIF3 deficiency led to higher expression of *MT-ND1*, *MT-ND2*, a trending increase of *MT-ND4*, and lower expression of *MT-ND3*, and *MT-CO3* (*Figure 4C*). In addition, we also found significantly reduced mitochondrial content in *MTIF3* knockout adipocytes (*Figure 4D*). Taken together, the above data suggest *MTIF3* knockout disrupts mitochondrial DNA-encoding gene and protein expression.

Next, we hypothesized the above observations could have originated from the insufficient MTIF3 supply during OXPHOS complex assembly (a role previously ascribed to MTIF3 [*Rudler et al., 2019*]). To test this, we used Blue Native-PAGE to examine OXPHOS complexes in mitochondria isolated from *MTIF3* knockout adipocytes. MTIF3 deficiency led to decreased complex III$_2$/IV$_2$ and IV$_1$, and a trending

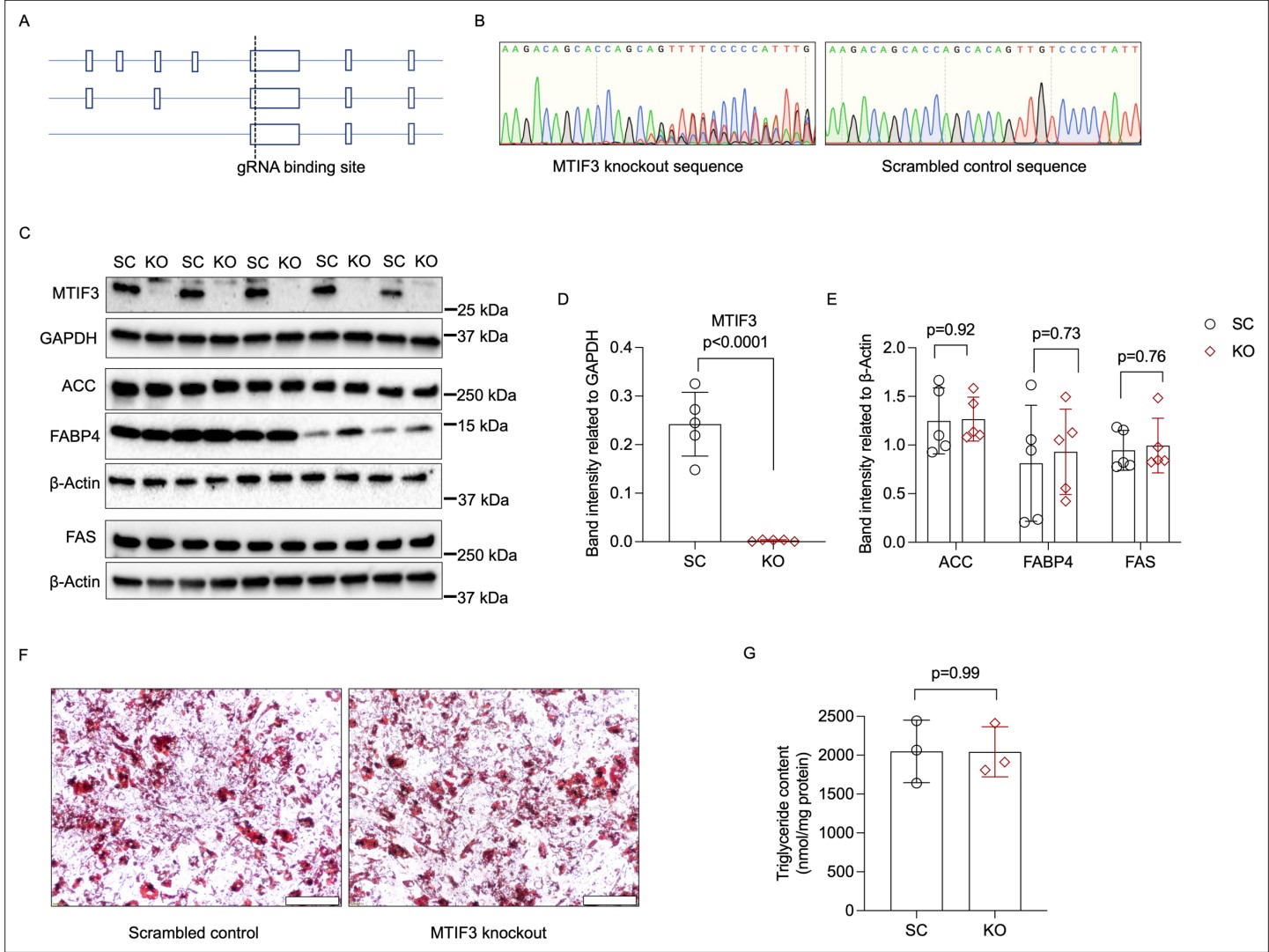

**Figure 3.** MTIF3 perturbation in mature adipocytes does not affect adipocyte-specific protein expression or total triglyceride content. (**A**) An illustration of Cas9-specific single guide RNA (sgRNA)-binding site in the exon expressed in all three *MTIF3* protein-encoding transcripts. (**B**) Representative Sanger sequencing of control and knockout hWAs mature adipocytes. (**C**) Immunoblots of adipocyte markers in scrambled control and *MTIF3* knockout adipocytes, n = 5 independent experiments. (**D**) Quantitative analysis of MTIF3 band densities in (**C**). (**E**) Quantitative analysis of ACC, FABP4, and FAS band densities in (**C**). (**F**) Representative Oil-red O staining images of control and MTIF3 knockout in hWAs mature adipocytes. Scale bar is 200 μm. (**G**) Total triglyceride content in scrambled control (SC) and *MTIF3* knockout (KO) cells. n = 3 independent experiments. Error bars show standard deviation in all plots. Statistical analysis was performed using two-tailed Student's *t*-test, p values are presented in each graph. Uncropped blot images for (**C**) and raw.scn data files can be found in *Figure 3—source data 1*.

The online version of this article includes the following source data and figure supplement(s) for figure 3:

**Source data 1.** Raw data files for western blots shown in *Figure 3C*.

**Figure supplement 1.** To test the effects of *MTIF3* knockout on adipogenic differentiation in hWAs-iCas9 cell line, we first induced *MTIF3* knockout in hWAs-iCas9 pre-adipocytes, then differentiated them using standard adipogenic differentiation cocktail.

**Figure supplement 1—source data 1.** Raw data files for western blots shown in *Figure 3—figure supplement 1A*.

decreased complex V/III$_2$ + IV$_1$ assembly. In contrast, OXPHOS complex II assembly was significantly increased in *MTIF3* knockout cells (*Figure 4E, F*). Interestingly, we also observed faster-migrating undefined bands in *MTIF3* knockout adipocytes (*Figure 4E*), which could be single chain proteins, or mistranslation or degradation products. Lastly, The OXPHOS complex I + III$_2$ + IV$_n$ appeared to be less abundant in *MTIF3* knockout mitochondria, although the bands appeared more diffuse and could not be quantified (*Figure 4E*).

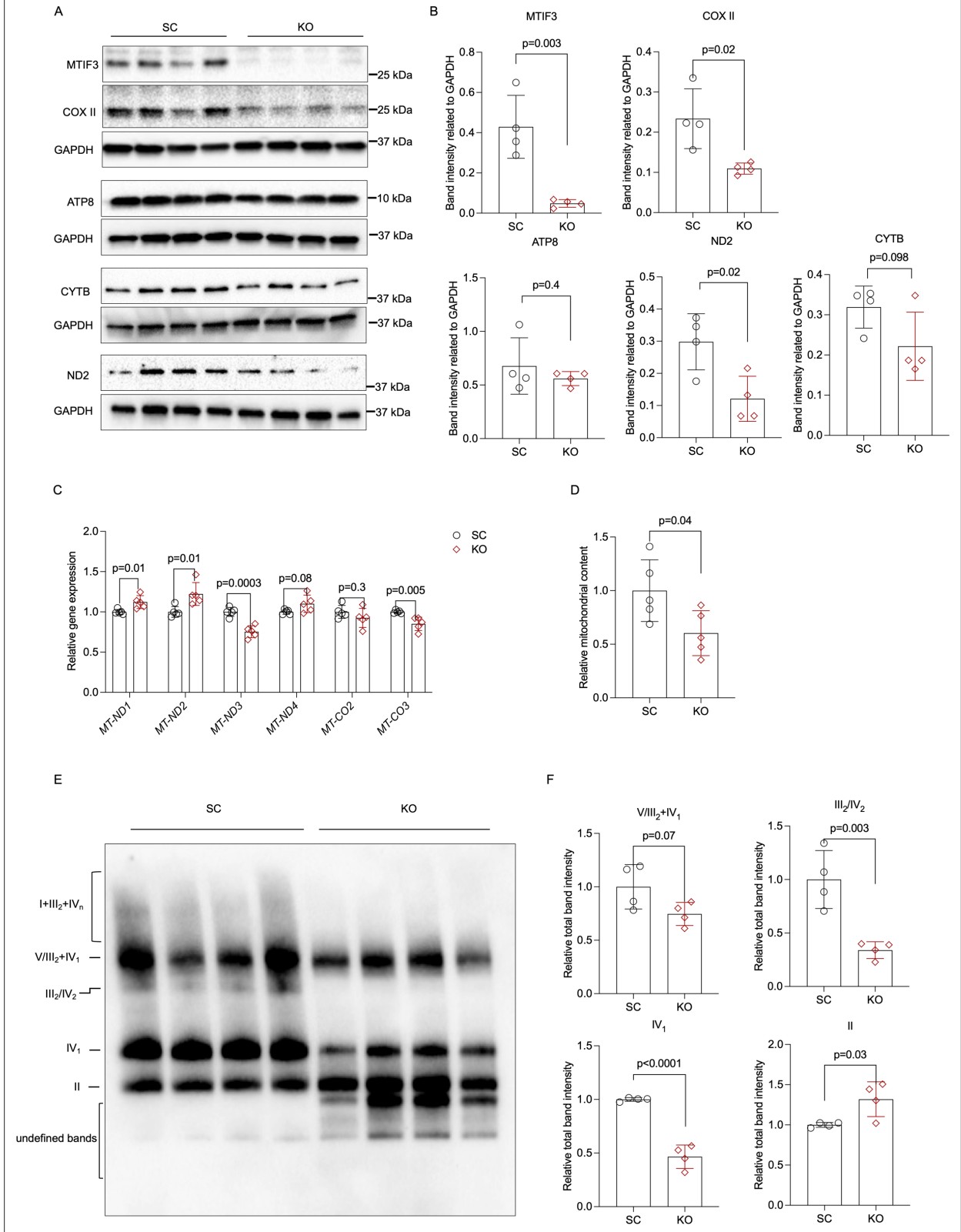

**Figure 4.** MTIF3 perturbation in mature adipocytes disrupts mitochondrial gene expression and OXPHOS complex assembly. (**A**) Immunoblots of mitochondrial genome-encoded proteins in scrambled control and *MTIF3* knockout adipocytes. (**B**) Quantitative analysis of band densities in (**A**). (**C**) qPCR for mitochondrial gene expression in scrambled control and *MTIF3* knockout adipocytes, *n* = 5 independent experiments. (**D**) Relative mitochondrial DNA content in scrambled control and *MTIF3* knockout adipocytes, *n* = 5 independent experiments. (**E**) Immunoblots of mitochondrial

*Figure 4 continued on next page*

*Figure 4 continued*

OXPHOS complex assembly after Blue Native-PAGE electrophoresis, *n* = 4 independent experiments. (**F**) Quantitative analysis of band densities in (**E**). Error bars show standard deviation in all plots. Statistical analysis was performed using two-tailed Student's *t*-test, p values are presented in each graph. Uncropped blot images for (**A**) and raw.scn data files can be found in *Figure 4—source data 1*. Uncropped blot images for (**E**) and raw.scn data files can be found in *Figure 4—source data 2*.

The online version of this article includes the following source data for figure 4:

**Source data 1.** Raw data files for western blots shown in *Figure 4A*.

**Source data 2.** Raw data files for western blots shown in *Figure 4E*.

### *MTIF3* knockout affects mitochondrial respiration in hWAs-iCas9 adipocytes

Having established the role of MTIF3 in adipocyte mitochondria OXPHOS complex assembly, and in mitochondrial gene expression, we then investigated the mitochondrial function in MTIF3-ablated differentiated hWAs adipocytes using Seahorse Mito Stress Test. Additionally, to avoid potential cofounders caused by the high glucose content in the differentiation medium, we adapted the cells to 1 g/l glucose growth medium for 3 days before running the assay. As shown in *Figure 3*, *MTIF3* knockout cells exhibited lower basal oxygen consumption rate (OCR), as well as lower ATP-forming capacity, the latter estimated by calculating OCR decrease after blocking ATP synthase with oligomycin (*Figure 5A–C*). *MTIF3* knockout cells also showed a trending decrease in maximal respiration OCR (*Figure 5D*). Furthermore, both *MTIF3* knockout and control cells, had comparable proton leak OCR, non-mitochondrial respiration OCR and coupling efficiency (*Figure 5E–G*).

### *MTIF3* knockout affects hWAs-iCas9 adipocyte endogenous fatty acid oxidation

Next, we used Seahorse to assess the endogenous fatty acid oxidation in *MTIF3* knockout versus control cells treated with etomoxir (an inhibitor of carnitine palmitoyl transferase). We found that MTIF3 ablation mimics the effect of etomoxir on basal endogenous fatty acid oxidation OCR. Furthermore, while etomoxir decreases basal fatty acid oxidation OCR in control cells, it does not markedly decrease it in *MTIF3* knockout cells (*Figure 6A, B*).

### *MTIF3* knockout affects hWAs-iCas9 adipocyte triglyceride content after glucose restriction challenge

To mimic the interactions between MTIF3 content and dietary intervention on weight change, we generated hypertrophic control and *MTIF3* knockout hWAs-iCas9 adipocytes and then used glucose-limited medium, not supplemented with free fatty acids (FFAs), to mimic energy restriction in vivo (schematic shown in *Figure 6C*). Triglyceride content decreased both in control and *MTIF3* knockout cells after 3 days of different levels of glucose restriction when compared with 25 mM glucose medium. Interestingly, a more extensive decrease in triglyceride content occurred in control cells cultured in 1 mM glucose medium (p = 0.053), and a similar trending decrease, albeit with higher coefficient of variation, occurred in 3 and 5 mM glucose medium (*Figure 6C*).

### *MTIF3* knockout does not affect lipolysis-mediated glycerol release in hWAs adipocytes

Owing to the effects of MTIF3 ablation on triglyceride content and on fatty acid oxidation, described above, we then examined the effects of *MTIF3* knockout on lipolysis. We measured basal, insulin-attenuated, and isoproterenol-stimulated glycerol release in differentiated hWAs cells. As shown in *Figure 6D*, in all three conditions, glycerol release in control and *MTIF3* knockout cells was comparable. In addition, basal glycerol release in glucose-restricted conditions was similar (*Figure 6—figure supplement 1*), and significantly reduced in low glucose versus high glucose assay medium.

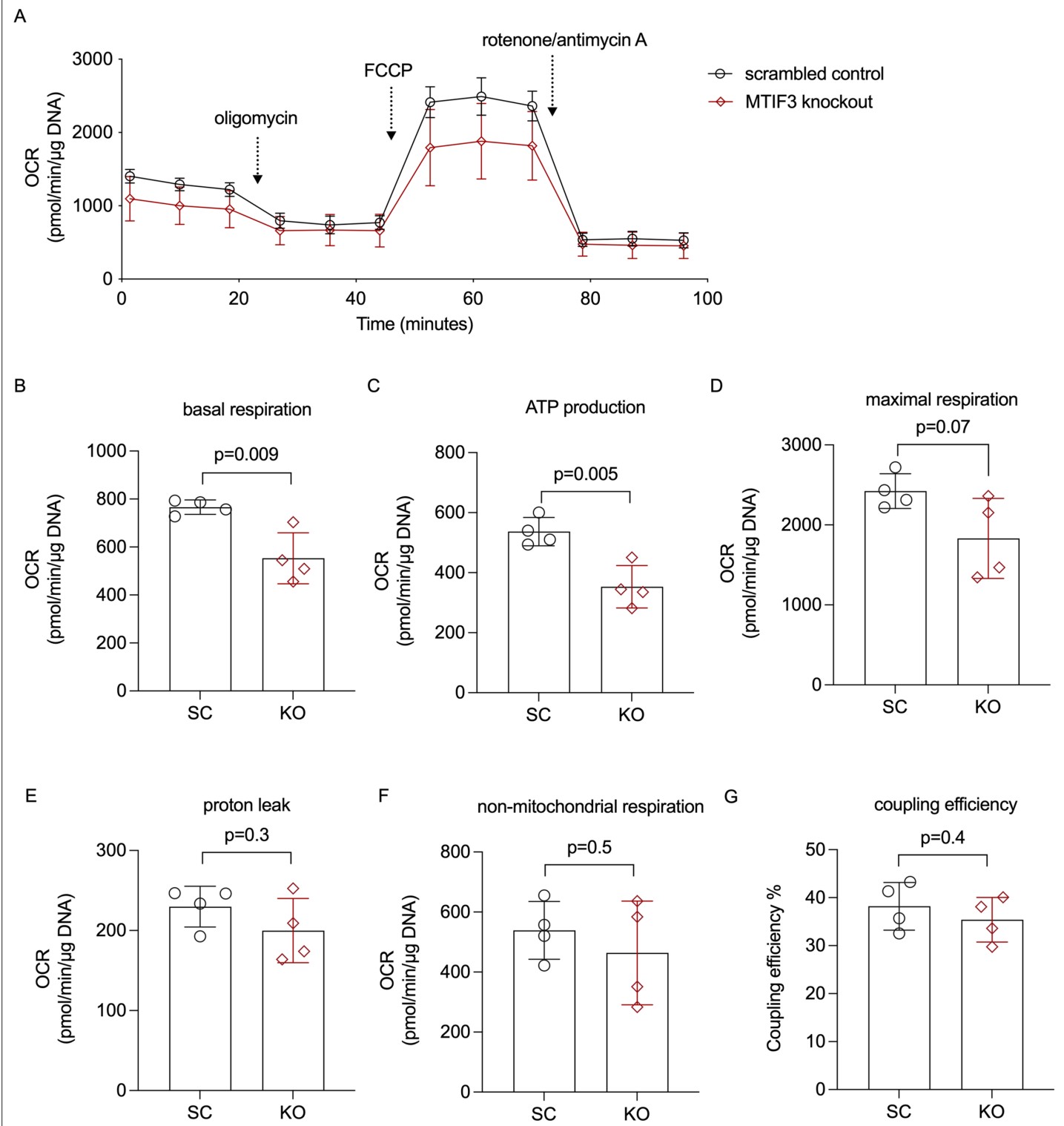

**Figure 5.** Cellular mitochondrial respiration in hWAs adipocytes. (**A**) The average oxygen consumption rate (OCR) traces during basal respiration, and after addition of oligomycin, FCCP, and rotenone/antimycin A. (**B**) Basal respiration OCR, $n$ = 4 different cell passages. (**C**) ATP production OCR, $n$ = 4 different cell passages. (**D**) Maximal respiration OCR, $n$ = 4 different cell passages. (**E**) Proton leak OCR, n = 4 different cell passages. (**F**) Non-mitochondrial respiration OCR, n = 4 different cell passages. (**G**) Coupling efficiency, $n$ = 4 different cell passages. Error bars show standard deviation. Statistical analyses were performed using paired Student's $t$-test in each condition, p values are presented in each graph.

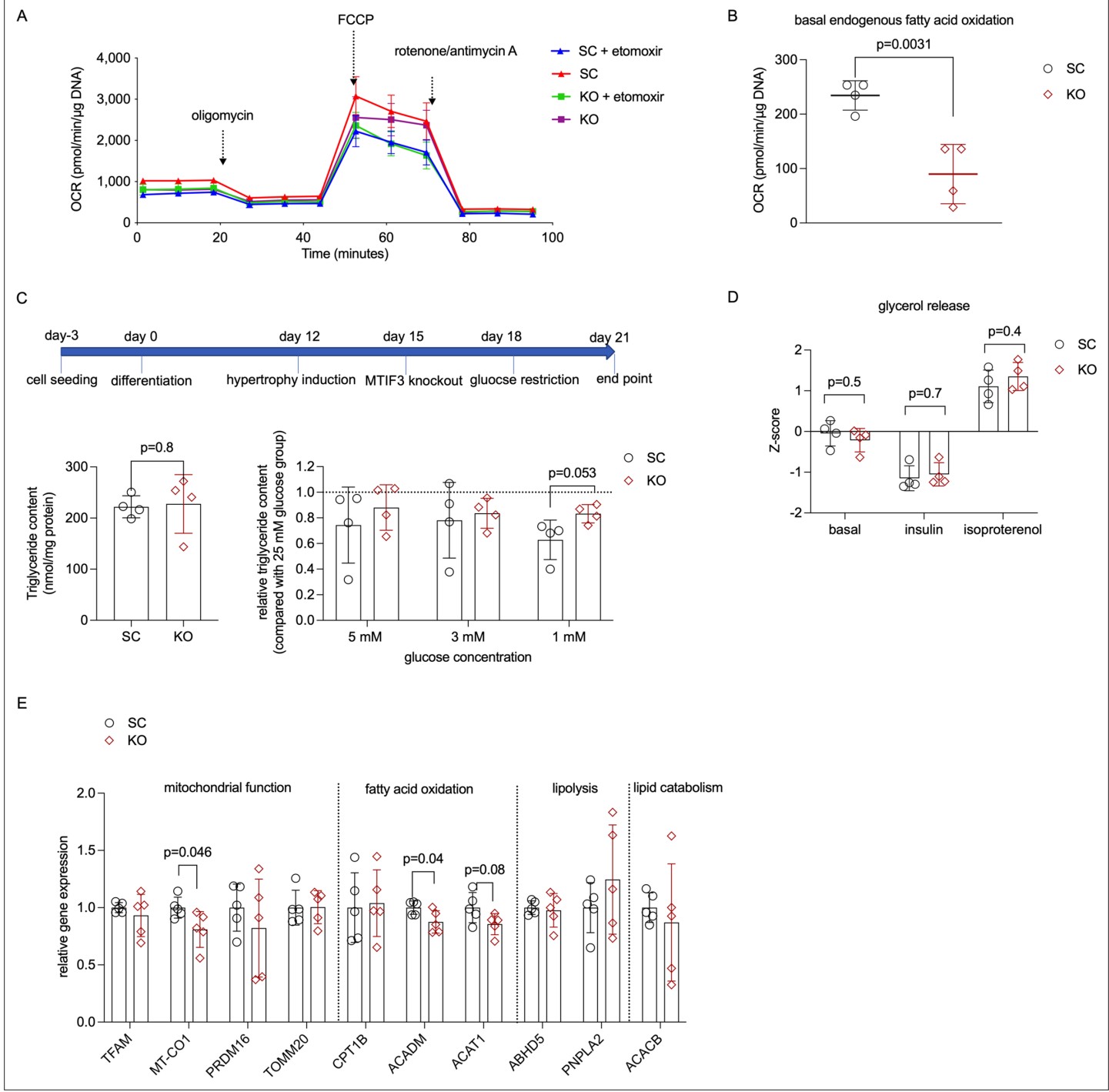

**Figure 6.** MTIF3 perturbation affects adipocyte fatty acid oxidation. (**A**) A representative Seahorse oxygen consumption rate (OCR) trace for endogenous fatty acid oxidation assay. *MTIF3* knockout and scrambled control adipocytes were treated with or without etomoxir for 15 min before the assay. Following the basal OCR measurement, oligomycin, FCCP (carbonyl cyanide-*p*-trifluoromethoxyphenylhydrazone), and rotenone + antimycin A were added sequentially to measure the detection of ATP production OCR, maximal respiration OCR and non-mitochondrial respiration OCR. (**B**) Basal endogenous fatty acid oxidation OCR in scrambled control (SC) and *MTIF3* knockout (KO) adipocytes, *n* = 4 independent experiments. (**C**) Upper panel: workflow of glucose restriction in differentiated adipocytes; Lower left panel: total triglyceride content in scrambled control (SC) and *MTIF3* knockout (KO) adipocytes in 25 mM glucose conditions; Lower right panel: triglyceride content in adipocytes cultured in glucose-restricted conditions (5, 3, and 1 mM) relative to adipocytes cultured in 25 mM glucose, *n* = 4 independent experiments. (**D**) Z-score-normalized data for glycerol release in scrambled control and *MTIF3* knockout adipocytes under basal, insulin-stimulated, and isoproterenol-stimulated conditions, *n* = 4 independent experiments.

*Figure 6 continued on next page*

*Figure 6 continued*

(**E**) qPCR for mitochondrial and adipocyte-related gene expression in scrambled control and *MTIF3* knockout adipocytes. Error bars show standard deviation in all plots. Statistical analysis was performed using two-tailed Student's *t*-test, p values are presented in each graph.

The online version of this article includes the following figure supplement(s) for figure 6:

**Figure supplement 1.** *MTIF3* knockout does not affect mature adipocyte glycerol release at either 25 or 1 mM glucose condition.

## *MTIF3* knockout affects mitochondrial function- and fatty acid oxidation-related gene expression

Next, we examined how MTIF3 ablation affects the gene expression programmes pertinent to mitochondrial function, fatty acid oxidation, lipolysis and lipid catabolism. As shown in *Figure 6E*, *MTIF3* knockout cells had decreased expression of the mitochondria-related *MT-CO1*, and the fatty acid oxidation-related *ACADM* and *ACAT1*, but unchanged expression of other genes involved in mitochondrial function and lipid metabolism (*TFAM*, *TOMM20*, *PRDM16*, *CPT1B*, *ABHD5*, *PNPLA2*, and *ACACB*).

## *MTIF3* knockout results in glucose level-depending alterations in metabolism

Considering the observed effect of MTIF3 ablation on mitochondrial function and fatty acid oxidation, we assessed the metabolite profile in *MTIF3* knockout cells. Using combined GC/MS and LC/MS metabolite profiling resulted in relative quantification of 110 metabolites. First, we analyzed metabolite profiles at a global level using PCA. The score plot reveals a clear systematic difference in the metabolite profile between cells in 25 mM glucose versus in glucose restriction (*Figure 7A*). Interestingly, differences between *MTIF3* knockout and control cells at 25 mM glucose are observed along principal component 1 (PC1), whereas differences between genotypes at glucose restriction are observed along PC2, suggesting the effect of MTIF3 ablation to depend on the calorie level. Next, to identify alterations in metabolite levels underlying this differential response, we analyzed data using orthogonal projections to latent structures discriminant analysis (OPLS-DA) separately at 25 mM glucose condition (two components $R2 = 0.82$, $Q2 = 0.66$) and at glucose restriction (two components, $R2 = 0.95$, $Q2 = 0.52$). These analyses revealed systematic differences between genotypes at both growth conditions (*Figure 7 B, C*). Next, to examine whether the differences between genotype depended on growth condition, we combined the correlations from the two OPLS-DA models into a shared and unique structures plot (*Figure 7D*). These analyses revealed levels of intermediates in cytosolic metabolic pathways connected to the glycolysis, such as glycerate 3-phosphate, glycerol 2-phosphate, UDP-*N*-acetylglucosamine, and ribose 5-phosphate, to be lower in *MTIF3* knockout cells at both glucose concentrations. Interestingly, levels of fatty acids, ranging from 9 to 17 carbons and including several odd-chain fatty acids, were lower in the *MTIF3* knockout cells only at 25 mM glucose condition. At glucose-restricted conditions, levels of both essential and non-essential amino acids were lower in the knockout. Finally, we analyzed data using two-way analysis of variance (ANOVA), incorporating glucose concentration and genotype, thereby providing information on effects at the individual metabolite level. These analyses revealed 18 and 20 significantly different metabolites between control and *MTIF3* knockout cells at 25 mM glucose condition and glucose-restricted conditions, respectively ($q < 0.05$). These included ribose 5-phosphate, glycerate 3-phosphate, glycerol 2-phosphate, and glycerol 3-phosphate (*Figure 7E*).

## Discussion

Excessive weight gain caused by dietary excess, and its effects on adipocyte lipid metabolism, can cause life-threatening disease (*Appleton et al., 2013*; *Denis and Obin, 2013*; *Chu et al., 2017*; *Lotta et al., 2018*). Findings from clinical trials (*Papandonatos et al., 2015*), a bariatric surgery case series (*Rasmussen-Torvik et al., 2015*) and epidemiological cohorts (*Nettleton et al., 2015*) showed the *MTIF3* variation modulates weight loss-promoting exposures on body weight (see also UK Biobank analysis in *Supplementary file 1b*). Here, we validated *MTIF3* rs1885988 C allele correlates with higher *MTIF3* expression in subcutaneous fat tissue, and our in vitro luciferase reporter assay and

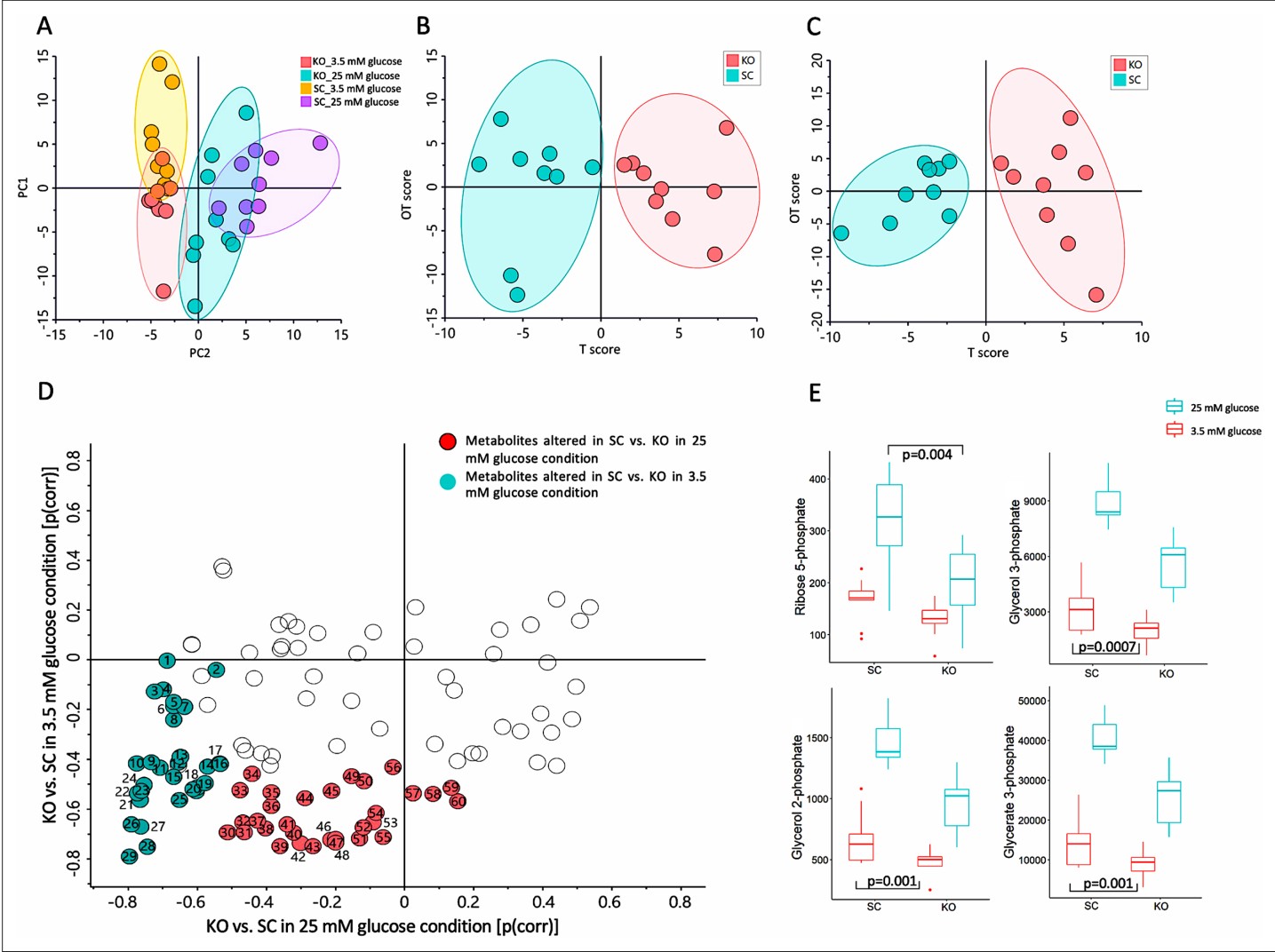

**Figure 7.** Mass spectrometry-based metabolomics data for control (SC) and *MTIF3* knockout (KO) cells in 25 mM glucose (NF, normal feeding) and 5 mM glucose (GR, glucose-restricted) conditions. (**A**) Principal component analysis (PCA) score plot displaying the discrimination between *MTIF3* knockout and control cells in normal and glucose-restricted conditions (PC1: 28%, PC2: 19%). (**B**) Orthogonal projections to latent structures discriminant analysis (OPLS-DA) score plot showing classification of *MTIF3* knockout and control cells in 25 mM glucose condition. (**C**) OPLS-DA score plots showing classification of *MTIF3* knockout and control cells in glucose-restricted condition. (**D**) Shared and unique structures (SUS) plot, based on OPLS-DA models in (**B, C**), showing glucose concentration-dependent differences between *MTIF3* knockout and control cells. (**E**) Box plots showing the abundance of some of the significantly altered metabolites in normal and *MTIF3* knockout cells in normal and glucose-restricted conditions. Statistical analysis was performed using two-way analysis of variance (ANOVA) test, p values are presented in each graph.

CRISPR-Cas9 genome editing results revealed that the tightly linked rs67785913 variant is likely to be the actual eQTL for *MTIF3* expression (*Figure 1*).

Pre-adipocyte cell lines have been used extensively for studying adipogenic differentiation (*Ruiz-Ojeda et al., 2016*), but their application for lipid metabolism studies has been limited, especially in the context of gene–environment interaction. This is largely due to the variation of differentiation capacity across cell passages (*Poulos et al., 2010*) and differential genetic effects on adipocyte differentiation (*Kamble et al., 2020*), which can alter baseline phenotypes in differentiated cells. Therefore, we established an inducible Cas9-expressing human pre-adipocyte cell line (hWAs-iCas9), which enabled us to generate gene knockout of interest in differentiated adipocytes, thus circumventing these limitations (*Figure 2*). Using the inducible knockout cell model, we then tested interactions between *MTIF3* and environmental changes (lifestyle mimetics). Our data reveal that *MTIF3* deficiency mediated disrupted mitochondrial respiration, probably as a consequence of decreased OXPHOS

complex assembly. This led to perturbed cellular functions, including reduced fatty acid oxidation and a trending increase in intracellular triglyceride content. These data indicate that *MTIF3* plays an important role in lipid metabolism in human adipocytes.

In adipose tissue, mitochondria play an essential role not only by ensuring ATP supply but also by triggering cellular signaling pathways that require reactive oxygen species generated by OXPHOS complexes I and III (*Tormos et al., 2011*). These, and complexes IV and V, are partially encoded by mitochondrial DNA (*Maechler and Wollheim, 2001*). The post-transcriptional rate-limiting translation step can be promoted by MTIF3, as it facilitates initiation complex formation on mitochondrial 55S ribosomes in the presence of MTIF2, fMet-tRNA, and poly(A,U,G) (*Bhargava and Spremulli, 2005*). Previous studies have shown that loss of MTIF3 results in an imbalanced assembly of OXPHOS complexes in muscle and heart in mouse (*Rudler et al., 2019*) and decreased translation of ATP6 mRNA in hepatocyte-like HepG2 cells (*Chicherin et al., 2020*). Our data show that in adipocytes MTIF3 deficiency results in lower content of mitochondrial DNA-encoding proteins and altered expression of several mitochondrial DNA-encoding genes. Furthermore, it leads to disassembly of mitochondrial respiration OXPHOS complex, along with impaired mitochondrial respiration rate (*Figures 4 and 5*). Altogether, our results suggest that MTIF3 vastly affects the mitochondrial electron transport chain.

Mitochondrial dysfunction in white adipose tissue has been frequently associated with obesity (*Kaaman et al., 2007*), with the presumed mechanisms being decreased fatty acid oxidation and ATP production (reviewed in *Heinonen et al., 2020*). We have found that one causal link connecting these factors may be the MTIF3 content (*Figures 5 and 6*). Furthermore, a previous study described an inverse relationship between mitochondrial capacity and weight change (*Jokinen et al., 2018*); thus, conceivably, any genetic variation-modulated change in *MTIF3* expression could influence a dietary intervention outcome. We attempted to test this hypothesis in vitro, exposing hypertrophic adipocytes to glucose restriction, thereby mimicking weight loss-promoting exposures in vivo. We found that MTIF3-deficient adipocytes exposed to glucose restriction challenge responded less through changed triglyceride content, indicating limited capacity for lipid metabolism under glucose restriction (*Figure 6*). Intriguingly, we did not observe altered lipolysis (glycerol release) in *MTIF3* knockouts, either in the presence or absence of insulin and isoproterenol, or during glucose restriction. Here, we speculate that *MTIF3* knockouts, having reduced fatty acid oxidation, re-esterify the released fatty acids to triglycerides, and thus retain more triglycerides during glucose restriction.

Mitochondrial function is robustly associated with insulin sensitivity (*Böhm et al., 2020*; *Pietiläinen et al., 2008*; *Heinonen et al., 2015*), both of which can be improved through lifestyle intervention aiming at weight loss (*Phielix et al., 2010*; *Toledo et al., 2007*; *Larson-Meyer et al., 2006*; *Civitarese et al., 2007*). Furthermore, impaired adipocyte differentiation derived from mitochondrial dysfunction in adipose tissue often associates with insulin resistance in humans and animal models (*Sakers et al., 2022*; *Zhu et al., 2022*). In our study, we did not observe impaired adipocyte differentiation due to MTIF3 deficiency or rs67785913 eQTL. These results, however, may not be directly translatable to in vivo conditions, in part because in vitro differentiation protocols employ relatively highly concentrated adipogenic compounds. Further in vivo studies are therefore needed to establish any link between MTIF3 content and insulin resistance. Similarly, long-term in vivo studies on the effect of altered *MTIF3* expression on body weight are warranted to illuminate the translatability of the short-term or low effect size in vitro experiments (e.g., *Figures 5 and 6*). We envisage that the MTIF3 effect on adipocyte metabolism, while not dramatic in some of the presented data, could translate into larger effect size over time in vivo. Here, MTIF3 effect on metabolism in other tissues (e.g., muscle) would further contribute to body weight regulation. Lastly, one should bear in mind that our in vitro data were generated in cells highly depleted of MTIF3, and the extent to which a more moderate MTIF3 deficiency in vivo (e.g., conferred by common genetic variants) influences adipogenic differentiation or long-term diet-induced weight loss is currently unknown.

Finally, the metabolomic analysis added another dimension to the role of MTIF3 in regulating adipocyte metabolism (*Figure 7*). The lower content of glycolysis intermediates and odd-chain fatty acids in *MTIF3* knockout adipocytes could indicate blunted lipogenesis, while the decreased essential and non-essential amino acid level in *MTIF3* knockout cells under glucose restriction could be an adaption to the lower energy supply owing to impaired mitochondrial function (*Hansson et al., 2004*). The metabolite data, and the earlier described lower fatty acid oxidation capacity of MTIF3-deficient cells, suggest that MTIF3 plays a vital role in triglyceride metabolism in adipocytes, and provides

further insight into the previously reported role of this protein on weight loss induced by dietary intervention (*Papandonatos et al., 2015*).

In summary, we experimentally demonstrated that the common genetic variant rs67785913 is a functional polymorphism that causes modulated *MTIF3* expression. Our functional genomics studies demonstrate that MTIF3 is essential for mitochondrial electron transport chain complex assembly, mitochondrial function, and lipid metabolism in human adipocyte cell lines. MTIF3 content may also influence adipocyte triglyceride content in conditions that mimic dietary restriction, reflecting a gene x environment interaction. Since our findings show that higher *MTIF3* content in adipocytes increases mitochondrial function, this helps explain the previously observed interaction between *MTIF3* locus and weight loss interventions. Furthermore, this suggests, although it remains to be demonstrated, that people who carry genetic variants that increase *MTIF3* expression (e.g., the rs67785913 CTCT allele) may benefit more from lifestyle interventions targeting weight loss.

Lastly, we established a novel, efficient, and simple method to generate gene knockouts in differentiated adipocyte cell lines, and the method should be suitable for generating other gene knockouts.

## Materials and methods
### Allele-specific *MTIF3* expression in subcutaneous adipose tissue
Data and plots presented in this manuscript for allele-specific *MTIF3* expression in subcutaneous adipose tissue were obtained from GTEx Portal (https://www.gtexportal.org/home/) on 10/5/2022.

### Cell culture
A human white pre-adipocyte cell line (hWAs) was previously isolated and immortalized from human subcutaneous white adipose tissue of a female subject, aged 56 with a BMI of 30.8. The generation and characterization of hWAs were described previously (*Xue et al., 2015*), and the cell line was kindly shared by Professor Yu-Hua Tseng (Joslin Diabetes Center, Harvard Medical School, USA). hWAS cells. For expansion, cells were cultured in 25 mM Dulbecco's Modified Eagle Medium (DMEM) with GlutaMAX (10566016, Thermo Fisher Scientific), 10% fetal bovine serum (FBS; HyClone, GE Healthcare, Uppsala, Sweden), and 1% (100 U/ml) penicillin/streptomycin (15140122, Thermo Fisher Scientific). The cells were passaged at 90% confluence, and tested negative for mycoplasma.

### DNA isolation and luciferase reporter assays
Genomic DNA was isolated from hWAs cells using DNeasy Blood and Tissue kit (69506, QIAGEN) according to the manufacturer's manual. To fine map the transcriptional regulatory regions in the *MTIF3* locus, we first identified the common genetic variants which were in tight linkage disequilibrium ($r^2 \geq$ 0.8) with the lead variant rs1885988 in HaploReg v4.1 (*Ward and Kellis, 2012*). The thus identified 31 SNPs were tiled down into 12 DNA segments of the *MTIF3* gene, as shown in *Supplementary file 1a*. These segments were then PCR amplified from hWAs DNA, all ranging from 700 to 1600 bp in size (depending on PCR primer design constraints), and with all SNP loci located several hundred bp from the ends of each fragment. The PCR primers were also designed to include flanking KpnI and EcoRV sites to allow cloning into the pGL4.23 minimal promoter luc2 luciferase reporter vector (Promega). For the reporter assays, hWAs were seeded into 96-well plates, and on the following day transfected with 95 ng of the pGL4.23 vectors and 5 ng pGL4.75 CMV-Renilla reporter vectors (for normalization), using Lipofectamine 3000 (Thermo Fisher Scientific), in technical duplicates. Two days after transfection, the luc2 and Renilla signals were detected using Dual-Glo Stop&Glo reagents (E2920, Promega). The averages of technical duplicates were used to calculate luc2:Renilla ratios, which were then *Z*-score normalized to allow statistical evaluation across four independent experiments.

### gRNAs and ssDNA design for CRISPR/Cas9 mediated editing of rs67785913 in hWAs cells
To edit the rs67785913 CT allele to the minor CTCT allele in hWAs cell genome, CRISPR/Cas9 D10A nickase (Alt-R S.p. Cas9 D10A Nickase V3, IDT) and two sgRNAs, and an ssDNA donor template were used. The sgRNA spacer sequences were: 5'-TTCAATAAGAAATTCCTCAA-3' and 5'-GAAGAAAA AGGGGGGGACACG-3'. The ssDNA sequence was 5'-TGTGGACTCGCAGTCTGCCCTTGAGGAA TTTCTTATTGAAGAAGAAAAAGAGGGGGGGACACGGGGGCCCAGACCCCCAGCACCCGGCTTTCGA

GCAGGCTC-3'. All oligonucleotides, sgRNAs, and ssDNA were purchased from Integrated DNA Technologies. The transfection was performed using Nucleofector 2b device (program A-033) (Lonza, Sweden) in nucleofector reagent L (Lonza, Sweden) mixed with $5 \times 10^5$ hWAs cells, 120 pmol Cas9 nickase, 104 pmol sgRNA, and 300 pmol ssDNA. To increase the homology directed repair (HDR) editing efficiency, cells were incubated at 32°C for 2 days in growth medium containing 30 µM HDR enhancer (Alt-R HDR Enhancer V2, IDT). Subsequently, cells were transferred to 37°C for 3 days. For single-cell cloning, the hWAs cells were seeded at low density (2 cells/well in a 96-well plate) and allowed to expand for 3 weeks. Then the genomic DNA was extracted using QuickExtract DNA Extraction Solution (Lucigen) from the apparent single-cell clonal populations. To identify the allele-edited homozygous clones, PCR was used to amplify the DNA fragment surrounding rs67785913 using primer pairs as below: Forward 5'–3' GATTTGCAGGTGAGCAGACA, Reverse 5'–3' ACTTGGAA ATGGCCAAGATG; the amplicon was then subjected to Sanger sequencing to confirm the DNA sequence of each clone.

## Generation of inducible CRISPR/Cas9-expressing hWAs cell line (hWAs-iCas9)

hWAs cells were first seeded at 80,000 cells/well in 6-well plates and transfected with 200 ng Super PiggyBac transposase (PB210PA-1, System Biosciences) and 500 ng pPB-rtTA-hCas9-puro-PB plasmid (kind gift from Dr. William Pu) (*Wang et al., 2017*) using Lipofectamine 3000 (Thermo Fisher Scientific). The plasmid carries a doxycycline-inducible promoter driving the expression of Cas9, and a puromycin resistance gene, all flanked by piggyBac transposon integration sequences. After 2 days, the transfected cells were selected and expanded for 3 weeks in growth medium with 1 µg/ml puromycin, to obtain cells with genomically integrated inducible Cas9 construct.

## Differentiation of hWAs-iCas9 pre-adipocytes into mature adipocytes

hWAs-iCas9 pre-adipocytes were seeded into 24- or 96-well plates at the density of 40,000 or 8000 cells/well, respectively. After 3 days, the cells reached confluency and were then incubated for 12 days with the differentiation cocktail, with medium changes every 3 days, as described before (*Shamsi and Tseng, 2017*). To increase the accumulation of lipid droplets, 30 µM FFA (Linoleic Acid-Oleic Acid-Albumin) (L9655, Sigma-Aldrich) was added to the differentiation medium (*Aprile et al., 2020*).

## CRISPR/Cas9 guide RNA design and off-targeting check

To generate *MTIF3* knockout adipocytes, guide RNA spacer sequence targeting *MTIF3* exon 5, expressed in all known *MTIF3* protein-encoding transcripts (as reported at https://www.ensembl.org), was selected. The spacer sequence was 5'-GCAATAGGGGACAACTGTGC-3', and full-length sgRNA was purchased from IDT. Furthermore, the hWAs genomic sequence surrounding the gRNA-binding site was amplified by PCR using the primers 5'-CCACTTGTCTTGGGGACAGT-3' and 5'-CTGGGAATG GTGGTTGAATC-3', then analyzed by Sanger sequencing to ensure sequence match between gRNA spacer and the intended target locus. The potential off-target sites were predicted using CRISPR-Cas9 guide RNA design checker (https://eu.idtdna.com), and the genomic regions surrounding the top 5 off-target sites were PCR amplified from the genomic DNA extracted from *MTIF3*-knockout and scramble control cells. The amplicons were then analyzed for any heteroduplexes generated by off-targeting using T7EI assay (IDT, Alt-R Genome Editing Detection Kit).

## sgRNA transfection and *MTIF3* knockout in mature adipocytes

After 12 days of differentiation, Cas9 expression was induced in mature adipocytes by adding 2 µg/ ml doxycycline to the growth medium. On the following day, 30 nM pre-designed sgRNA was delivered into the cells using Lipofectamine RNAiMAX (13778075, Thermo Fisher Scientific) according to the manufacturer's protocol; in parallel, 30 nM negative control crRNA (1072544, IDT) was used to transfect the scrambled control cells. One day post-transfection, cells were washed with phosphate-buffered saline (PBS) and incubated in normal growth medium for at least 3 days before functional assays carried out.

## Oil-red O staining

After MTIF3 knockout, the differentiated white adipocytes were washed twice with PBS and fixed for 10–20 min with 4% buffered formalin at room temperature. The cells were then stained with Oil-red O solution for 30 min at room temperature, followed by five washes with distilled water. The stained cells were visualized using light microscopy.

## Glucose restriction challenge for hWAs-iCas9 adipocytes

The hWAs-iCas9 adipocytes were firstly differentiated to mature adipocytes as described above (in FFA-supplemented medium), then the *MTIF3* knockouts and scrambled controls were generated, also as described above. After 3 days, the mature adipocytes were incubated in DMEM medium (11966025, Thermo Fisher Scientific) without FFA, and supplemented with different glucose concentrations (5, 3, and 1 mM) for the glucose restriction test. The cells were then incubated for 3 days, and the triglyceride content was determined as described above.

## RNA isolation and qPCR gene expression assays

Total RNA was extracted from cells using RNeasy plus Kit (74034, QIAGEN) together with Qiazol reagent (79306, QIAGEN). RNA purity was assessed using Nanodrop (Nanodrop, Wilmington, USA), and cDNA was synthesized using SuperScript IV VILO Master Mix (11756500, Thermo Fisher Scientific). Then, RT-qPCR was performed on ViiA7 qRT-PCR system (PE Applied Biosystems, Foster City, CA, USA), using predesigned Taqman assays following the manufacturer's instructions. The Taqman assays (Thermo Fisher Scientific, Uppsala, Sweden) were: *MTIF3* (Hs00794538_m1), *GTF3A* (Hs00157851_m1), *ADIPOQ* (Hs00977214_m1), *PPARG* (Hs01115513_m1), *CEBPA* (Hs00269972_s1), *SREBF1* (Hs02561944_s1), *FASN* (Hs01005622_m1), *TFAM* (Hs01073348_g1), *MT-CO1* (Hs02596864_g1), *PRDM16* (Hs00223161_m1), *TOMM20* (Hs03276810_g1), *CPT1B* (Hs00189258_m1), *ACADM* (*Hs00936584_m1*), *ACAT1* (Hs00608002_m1), *ABHD5* (Hs01104373_m1), *PNP1A2* (Hs00386101_m1), *ACACB* (Hs01565914_m1), *MT-ND1* (Hs02596873_s1), *MT-ND2* (Hs02596874_g1), *MT-ND3* (Hs02596875_s1), *MT-ND4* (Hs02596876_g1), *MT-CO2* (Hs02596865_g1), *MT-CO3* (Hs02596866_g1), *HPRT-1* (Hs99999909_m1), *TBP* (Hs00427620_m1), and *RPL13A* (Hs03043885_g1). The relative gene expression was calculated using the delta Ct method, and the target gene expression was normalized to the mean Ct of three reference genes *HPRT-1*, *TBP*, and *RPL13A*.

## Western blotting

Cells were washed twice with ice-cold PBS and lysed in 1% sodium dodecyl sulfate buffer for 10 min, then passed through a QIAshredder (79654, QIAGEN) and centrifuged for 15 min at 14,000 × *g*. The supernatant was subsequently collected and protein concentration quantified using the BCA assays (23225, Thermo Fisher Scientific). To assess target protein expression, 10 µg lysates were loaded into 4–20% Mini-PROTEAN TGX Stain-Free Protein Gels (Bio-Rad Laboratories AB, Solna, Sweden) and separated, followed by transfer of polyvinylidene difluoride (PVDF) membranes (1704156, Bio-Rad Laboratories AB). After blocking in 5% bovine serum albumin (BSA) solution for 1 hr, the membranes were incubated with primary antibodies against MTIF3 (14219-1-AP, Proteintech), OXPHOS complex (45-8199, Thermo Fisher Scientific), FABP4, ACC, FAS (12589, Cell Signalling Technology), ATP8 (26723-1-AP, Proteintech), ND2 (19704-1-AP, Proteintech), CYTB (55090-1-AP, Proteintech), and corresponding horseradish peroxidase (HRP)-conjugated secondary antibodies (anti-mouse IgG, Cell Signalling Technology; anti-rabbit IgG, Cell Signaling Technology). TBS with 0.1% (vol/vol) Tween-20 was used for membrane washing, and TBS with 2% BSA was used for antibody incubation. To visualize the blots, Clarity western ECL substrate was added to the membrane and a CCD camera used to acquire images and Image Lab software (Bio-Rad Laboratories AB, Solna, Sweden) were used to develop the images. ImageJ software was used to quantify the protein bands. After detection of the protein targets, the membranes were stripped using Restore Western Blot Stripping Buffer (21059, Thermo Fisher Scientific) and blotted using anti-β-Actin antibody (4967, Cell Signaling Technology) or anti-GAPDH antibody (ab37168, Abcam).

## Blue Native polyacrylamide gel electrophoresis and immunoblotting

Differentiated scrambled control and *MTIF3* knockout hWAs-iCas9 cells were adapted to 5.5 mM glucose growth medium for 3 days to mimic the physiological glucose concentration. The Blue Native

polyacrylamide gel electrophoresis (BN-PAGE) was performed as described previously (*Singh and Duchen, 2022*). Briefly, mitochondria were isolated using Mitochondria Isolation Kit (89874, Thermo Fisher Scientific). NativePAGE Sample Prep Kit (BN2008, Invitrogen) was then used for mitochondrial protein extraction and BN-PAGE sample preparation. For native gel electrophoresis, 20 µg mitochondrial protein was loaded to precast 3–12% gradient Blue Native gels (BN1001, Invitrogen) and separated according to the manufacturer's instructions. The proteins were then electroblotted onto PVDF membrane and probed with anti-OXPHOS antibody cocktail (45-8199, Thermo Fisher Scientific) and a corresponding secondary antibody. The blots were then imaged and analyzed as described above.

## Relative mitochondrial content measurement

To examine the effects of *MTIF3* knockout on mitochondrial biogenesis in white adipocytes, relative amount of mtDNA was quantified using a qPCR-based method described previously (*Ajaz et al., 2015*). Briefly, total DNA was extracted and quantified using QIAamp DNA Mini Kit (catalogue number: 56304, QIAGEN) from scrambled control and *MTIF3* knockout cells. For qPCR, equal amounts of total DNA from each sample were mixed with SYBR Green master mix (catalogue number: A25742, Thermo Fisher Scientific) and with primers targeting mitochondrial and nuclear genes, then the samples were run on ViiA7 qRT-PCR system (PE Applied Biosystems, Foster City, CA, USA). The relative mtDNA content was calculated as ΔCt (Ct of nuclear target − Ct of mitochondrial target).

## Mitochondrial function in *MTIF3* knockout adipocytes

To directly assess the effects of *MTIF3* on mitochondrial respiration in adipocytes we used the Seahorse XF (Seahorse Bioscience, North Billerica, MA) to measure cellular respiration OCR under different conditions. hWAs-iCas9 cells were seeded at 8000 cells/well in a Seahorse 24-well plate, then differentiated and induced for *MTIF3* knockout or with scrambled control, as described above. Then, cells were adapted in 1 g/l growth medium (31885049, Thermo Fisher Scientific) for 3 days. Mitochondrial function was then assessed using the Seahorse XF-24 instrument according to a protocol optimized for the adipocyte cell line. Briefly, to measure OCR independent of oxidative phosphorylation, 2 µM oligomycin (O4876, Sigma-Aldrich) was added to the cells. Subsequently, 2 µM FCCP (carbonyl cyanide-*p*-trifluoromethoxyphenylhydrazone) (C2920, Sigma-Aldrich) and 5 µM respiratory chain inhibitors: rotenone (R8875, Sigma-Aldrich) and antimycin A (A8674, Sigma-Aldrich) were added to measure maximal respiration and basal rates of non-mitochondrial respiration. Cells were then frozen at −80°C for at least 4 hr, then the plate was dried, and DNA was extracted with CyQUANT Cell Lysis Buffer (C7027, Thermo Fisher Scientific). Total DNA was then quantified by Quant-iT PicoGreen dsDNA Assay Kit (P7589, Thermo Fisher Scientific) against a lambda DNA-generated standard curve.

## Endogenous long chain fatty acid oxidation in adipocytes

The Seahorse mitochondrial analyzer was used to test the effects of MTIF3 loss on endogenous long chain fatty acid oxidation in adipocytes. Prior to the assay, adipocytes were incubated overnight with substrate-limited medium: DMEM (A14430, Thermo Fisher Scientific); 0.5 mM glucose (103577-100, Angilent); 1.0 mM glutamine (103579-100, Angilent); 0.5 mM carnitine (C0283, Sigma-Aldrich); 1% FBS (SV30160.03, HyClone). On day of the assay, the substrate-limited medium was replaced with FAO assay medium: 1× Krebs-Henseleit Buffer (KHB) was supplemented with 2.5 mM glucose, 0.5 mM carnitine, and 5 mM N-2-hydroxyethylpiperazine-N-2-ethane sulfonic acid (HEPES), and the pH was adjusted to pH 7.4 with NaOH. The cells were then treated for 15 min with either 40 µM etomoxir (E1905, Sigma-Aldrich) or only with the solvent (dimethyl sulfoxide, DMSO). Etomoxir inhibits carnitine palmitoyltransferase (CPT)-1 and diglyceride acyltransferase (DGAT) activity in mitochondria, and thus inhibits mitochondrial fatty acid oxidation (*Xu et al., 2003*; *Griesel et al., 2010*). The OCR was then measured as described above.

## Mass spectrometry-based metabolite profiling

The mature hWAs-iCas9 adipocytes were firstly induced for *MTIF3* knockout, followed by glucose restriction challenge as described above. The cells were quenched on dry ice and metabolites were extracted using a previously optimized protocol (*Danielsson et al., 2010*).

For analysis of low molecular weight metabolites, extracts were reconstituted in 100 µl of MeOH/water (8/2, vol/vol) and 60 µl was transferred to new Eppendorf tubes and evaporated to dryness

using a miVac concentrator (SP Scientific, NY) for 3 hr at 30°C. Dried samples were methoximated using 20 µl of methoxyamine hydrochloride in pyridine (Thermo Scientific, MA) by shaking at 3000 rpm for 30 min at room temperature (VWR, PA). Afterward, 20 µl of *N*-methyl-*N*-(trimethylsilyl) trifluoro-acetamide (MSTFA) + 1% trimethylsilyl chloride (Thermo Scientific, MA) was added to each sample and shaken at 3000 rpm at room temperature for 1 hr. Samples were transferred to glass vials and immediately analyzed using an Agilent 6890 gas chromatograph connected to an Agilent 5975CL VL MSD mass spectrometer controlled by MassHunter Workstation software 10.0 (Agilent, Atlanta, GA). One µl sample was injected at 270°C on an HP-5MS column (30 m length, 250 µm ID, 0.25 µm phase thickness). with a helium gas flow rate of 1 ml/min and a temperature gradient starting at 70°C for 2 min, increasing 15°C/min to 320°C and held for 2 min. Data were acquired using electron ionization at 70 eV in either full scan (50–550 *m/z*) or single ion monitoring mode. The MS-DIAL version 4.7 was used for raw peak extraction, peak alignment, deconvolution, peak annotation, and integration of peaks.

Amino acids and free fatty acids were chemically derivatized and analyzed using a previously described method (*Meng et al., 2021*). Briefly, 40 µl of the samples were mixed with 20 µl of 3-nitrophenylhydrazine (3-NPH) (Sigma-Aldrich, MO), followed by addition of 20 µl of 1-ethyl-3 -(3-dimethylaminopropyl)carbodiimide hydrochloride (EDC) (Thermo Scientific, MA) and shaking at 3000 rpm at room temperature for 1 hr. Samples were analyzed using an Agilent 1260 ultra-performance liquid chromatograph coupled with an Agilent 6495 tandem mass spectrometer and controlled by MassHunter version 8.0 (Agilent Technologies, CA). Three µl sample was injected on an Agilent Eclipse RRHD C18 column (2.1 × 150 mm, 1.8 µm) (Agilent Technologies, CA) with a flow rate of 0.6 ml/min and a column oven temperature of 50°C. The mobile phases A and B were 0.1% formic acid (Fisher Chemical, Prague, Czech Republic) in Milli-Q water (Merck, Millipore, MO) and acetonitrile (VWR, Paris, France), respectively. Gradient elution was performed as follows: held at 5% B from 0 to 1 min, changed linearly to 90% B in 10 min, changed from 90% B to 100% B in 13 min, held at 100% B for 2 min, returned to 5% B (initial condition) in 0.1 min, and held at 5% B for 2 min. Analyses were conducted in negative electrospray ionization mode (ESI) mode with the nebulizer gas pressure set at 20 psi, ion capillary voltage at 2500 V, gas temperature at 150°C, and sheath gas temperature at 250°C. Data were recorded in multiple reaction monitoring (MRM) mode, with two transitions for each analyte.

## Lipolysis quantification in differentiated hWAs cells

Differentiated scrambled control or *MTIF3* knockout cells were washed twice with PBS and then incubated with DMEM containing 2% free fatty acid-free BSA for 2 hr. For the insulin or isoproterenol-stimulated lipolysis, 100 nM insulin (I2643, Sigma-Aldrich) or 10 µM isoproterenol (1351005, Sigma-Aldrich) was added in the medium separately. After the incubation, the medium was collected, and the glycerol content was measured using Glycerol-Glo Assay (J3150, Promega).

## Total triglyceride measurement

Triglyceride-Glo Assay kit (J3161, Promega) was used to quantify total triglyceride content in scrambled control or *MTIF3* knockout cells cultured either in 25 mM glucose or glucose restriction medium. Briefly, cells were collected in 50 µl kit lysis buffer at room temperature for 1 hr. Then 2 µl lysate was mixed with 8 µl glycerol lysis solution with lipase, and incubated at 37°C for 30 min. Subsequently, 10 µl glycerol solution was mixed with 10 µl glycerol detection solution supplemented with reductase substrate and kinetic enhancer, and transferred into a 384-well plate. After 1-hr incubation at room temperature, the luminescence was detected using CLARIOstar plate reader (BMG Labtech, Germany), and the triglyceride concentration was calculated using a standard curve generated from glycerol standards and normalized to total protein measured using BCA assays (23227, Thermo Fisher Scientific).

## Statistics

For each assay, the number of biological and technical replicates, standard deviation and statistical significance are reported in the figure legends. Hypothesis tests were performed using two-tailed Student's *t*-test, one-way ANOVA, or paired *t*-test. A nominal p value of <0.05 was considered statistically significant. All analyses were undertaken using Prism GraphPad 9.0 software (La Jolla California,

USA), SIMCA 17.0 (Sartorius Stedim Data Analytics, Malmö, Sweden), Rstudio 1.4, and Microsoft Excel 365. For the metabolome data, ANOVA (aov) was performed in R with genotype and glucose concentration as independent variables with Tukey's test post hoc (TukeyHSD). Significance was defined as $q < 0.05$ using multiple testing adjustment according to the false discovery rate (p.adjust).

## Acknowledgements

We thank Jennifer Doudna (UC Berkeley) for initial support with concepts relating to using CRISPR in in vitro studies of gene–environment interactions. We also thank Yu-Hua Tseng in Joslin Diabetes Centre for providing the hWAs cells.

This work was supported by China Scholarship Council (201708420158 to Mi Huang); the European Commission (ERC-2015-CoG-681742 NASCENT to PWF), and Swedish Research Council (Distinguish Young Research Reward in Medicine) (to PWF), LUDC-IRC and The Swedish Research Council (to HM), and by The Albert Påhlsson Foundation (to SK).

## Additional information

### Funding

| Funder | Grant reference number | Author |
| --- | --- | --- |
| European Research Council | ERC-2015-CoG -681742 NASCENT | Paul W Franks |
| Vetenskapsrådet | | Hindrik Mulder |
| LUDC-IRC | | Hindrik Mulder |
| China Scholarship Council | 201708420158 | Mi Huang |
| Albert Påhlsson Foundation | | Sebastian Kalamajski |
| The Crafoord Foundation | 20210571 | Sebastian Kalamajski |

The funders had no role in study design, data collection, and interpretation, or the decision to submit the work for publication.

### Author contributions

Mi Huang, Formal analysis, Investigation, Methodology, Writing – original draft; Daniel Coral, Formal analysis, Investigation, Writing – original draft; Hamidreza Ardalani, Investigation, Methodology, Writing – original draft; Peter Spegel, Formal analysis, Investigation; Alham Saadat, Conceptualization, Resources, Supervision; Melina Claussnitzer, Conceptualization, Resources, Funding acquisition; Hindrik Mulder, Resources, Funding acquisition; Paul W Franks, Conceptualization, Supervision, Funding acquisition, Writing – original draft, Project administration; Sebastian Kalamajski, Conceptualization, Formal analysis, Supervision, Funding acquisition, Investigation, Methodology, Writing – original draft, Project administration

### Author ORCIDs

Mi Huang http://orcid.org/0000-0001-6938-5322
Hindrik Mulder http://orcid.org/0000-0002-6593-8417
Sebastian Kalamajski http://orcid.org/0000-0002-6600-9302

### Decision letter and Author response

Decision letter https://doi.org/10.7554/eLife.84168.sa1
Author response https://doi.org/10.7554/eLife.84168.sa2

## Additional files

### Supplementary files

• Supplementary file 1. Design of fine mapping luciferase reporter assays, and association of *MTIF3* locus with adiposity traits in UK Biobank. (a) Thirty-one SNPs in tight linkage disequilibrium ($r^2 \geq 0.8$) with the lead variant rs1885988 tiled down into 12 DNA segments of the *MTIF3* gene for luciferase reporter assay. To fine map the transcriptional regulatory regions in the *MTIF3* locus, we first identified the common genetic variants which were in tight linkage disequilibrium ($r^2 \geq 0.8$) with the lead variant rs1885988 in HaploReg v4.1. The identified 31 SNPs were tiled down into 12 DNA segments of the *MTIF3* gene depending on PCR primer design constraints. (b) SNPs in *MTIF3* locus associated with body mass index (BMI), whole-body fat mass and arm fat mass (right). We checked the rapid GWAS analysis results from 337,000 samples in the UK Biobank, which were made available by Benjamin Neale's lab and visualized in Oxford BIG browser, we found SNPs in *MTIF3* locus showed nominal associations with body weight-related traits including BMI, whole-body fat mass and arm fat mass (right).

• MDAR checklist

### Data availability

All data generated or analyzed in this study are included in the figures and the source data files. Source data files are provided for Figures 3 and 4, and for Figure 3—figure supplement 1.

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

# Appendix 1

## Appendix 1—key resources table

| Reagent type (species) or resource | Designation | Source or reference | Identifiers | Additional information |
|---|---|---|---|---|
| Gene (*Homo sapiens*) | MTIF3 | UCSC Genome Browser | GRCh38/hg38 | |
| Cell line (*Homo sapiens*) | hWAs | Tseng laboratory at Joslin Diabetes Center | *Xue et al., 2015* | |
| Cell line (*Homo sapiens*) | hWAs-iCas9 | This paper | | Cell line maintained at Lund University Diabetes Center |
| Antibody | Anti-MTIF3 (rabbit polyclonal antibody) | Proteintech | Cat: 14219-1-AP | WB (1:2000) |
| Antibody | Anti-OXPHOS antibody cocktail (mouse polyclonal antibody) | Thermo Fisher Scientific | Cat: 45-8199 | WB (1:1000) |
| Antibody | Anti-FABP4 (rabbit polyclonal antibody) | Cell Signalling Technology | Cat: 12589 | WB (1:1000) |
| Antibody | Anti-ACC rabbit polyclonal antibody | Cell Signalling Technology | Cat: 12589 | WB (1:1000) |
| Antibody | Anti-FAS rabbit polyclonal antibody | Cell Signalling Technology | Cat: 12589 | WB (1:1000) |
| Antibody | Anti-ATP8 (rabbit polyclonal antibody) | Proteintech | Cat: 26723-1-AP | WB (1:2000) |
| Antibody | Anti-ND2 (rabbit polyclonal antibody) | Proteintech | Cat: 19704-1-AP | WB (1:2000) |
| Antibody | Anti-CYTB (rabbit polyclonal antibody) | Proteintech | Cat: 55090-1-AP | WB (1:2000) |
| Antibody | Anti-β-Actin (rabbit polyclonal antibody) | Cell Signaling Technology | Cat: #4967 | WB (1:10,000) |
| Antibody | Anti-GAPDH (rabbit polyclonal antibody) | Abcam | Cat: ab37168 | WB (1:10,000) |
| Antibody | Anti-rabbit IgG, HRP-linked Antibody (goat polyclonal antibody) | Cell Signaling Technology | Cat: #7074 | WB (1:10,000) |
| Antibody | Anti-mouse IgG, HRP-linked Antibody (horse polyclonal antibody) | Cell Signaling Technology | Cat: #7076 | WB (1:10,000) |
| Recombinant DNA reagent | Super PiggyBac transposase (plasmid) | System Biosciences | PB210PA-1 | |
| Recombinant DNA reagent | pGL4.23 vectors | Promega | E8411 | |
| Recombinant DNA reagent | pGL4.75 CMV-Renilla reporter vectors | Promega | E6931 | |
| Recombinant DNA reagent | pPB-rtTA-hCas9-puro-PB plasmid | doi:10.1038/nprot.2016.152 | | |
| Sequence-based reagent | PCR primer (Forward) for rs67785913 genotyping | IDT | | 5'–3': GATTTGCAGGTGAGCAGACA |
| Sequence-based reagent | PCR primer (Reverse) for rs67785913 genotyping | IDT | | 5'–3': ACTTGGAAATGGCCAAGATG |
| Sequence-based reagent | sgRNA for rs67785913 editing | IDT | | Spacer sequence: 5'-TTCAATAAGAAATTCCTCAA-3' |
| Sequence-based reagent | sgRNA for rs67785913 editing | IDT | | Spacer sequence: 5'-GAAGAAAAAGGGGGGACACG-3' |
| Sequence-based reagent | Donor template for rs67785913 editing | IDT | | ssDNA sequence: 5"TGTGGACTCGCAGTCTGCCCTTGAGGAATTTCTTATTGAAGAAGAAAAAGAGGGGGGACACGGGGCCCAGACCCCCAGCACCCGGCTTTCGAGCAGGCTC-3' |
| Sequence-based reagent | sgRNA against MTIF3 | IDT | Design ID: Hs.Cas9.MTIF3.1.AB | Spacer sequence: 5'-GCAATAGGGGACAACTGTGC-3' |
| Sequence-based reagent | Taqman assay for *MTIF3* | Thermo Fisher Scientific | Hs00794538_m1 | |

*Appendix 1 Continued on next page*

*Appendix 1 Continued*

| Reagent type (species) or resource | Designation | Source or reference | Identifiers | Additional information |
|---|---|---|---|---|
| Sequence-based reagent | Taqman assay for GTF3A | Thermo Fisher Scientific | Hs00157851_m1 | |
| Sequence-based reagent | Taqman assay for ADIPOQ | Thermo Fisher Scientific | Hs00977214_m1 | |
| Sequence-based reagent | Taqman assay for PPARG | Thermo Fisher Scientific | Hs01115513_m1 | |
| Sequence-based reagent | Taqman assay for CEBPA | Thermo Fisher Scientific | Hs00269972_s1 | |
| Sequence-based reagent | Taqman assay for SREBF1 | Thermo Fisher Scientific | Hs02561944_s1 | |
| Sequence-based reagent | Taqman assay for FASN | Thermo Fisher Scientific | Hs01005622_m1 | |
| Sequence-based reagent | Taqman assay for TFAM | Thermo Fisher Scientific | Hs01073348_g1 | |
| Sequence-based reagent | Taqman assay for MT-CO1 | Thermo Fisher Scientific | Hs02596864_g1 | |
| Sequence-based reagent | Taqman assay for PRDM16 | Thermo Fisher Scientific | Hs00223161_m1 | |
| Sequence-based reagent | Taqman assay for TOMM20 | Thermo Fisher Scientific | Hs03276810_g1 | |
| Sequence-based reagent | Taqman assay for CPT1B | Thermo Fisher Scientific | Hs00189258_m1 | |
| Sequence-based reagent | Taqman assay for ACADM | Thermo Fisher Scientific | *Hs00936584_m1* | |
| Sequence-based reagent | Taqman assay for ACAT1 | Thermo Fisher Scientific | Hs00608002_m1 | |
| Sequence-based reagent | Taqman assay for ABHD5 | Thermo Fisher Scientific | Hs01104373_m1 | |
| Sequence-based reagent | Taqman assay for PNP1A2 | Thermo Fisher Scientific | Hs00386101_m1 | |
| Sequence-based reagent | Taqman assay for ACACB | Thermo Fisher Scientific | Hs01565914_m1 | |
| Sequence-based reagent | Taqman assay for MT-ND1 | Thermo Fisher Scientific | Hs02596873_s1 | |
| Sequence-based reagent | Taqman assay for MT-ND2 | Thermo Fisher Scientific | Hs02596874_g1 | |
| Sequence-based reagent | Taqman assay for MT-ND3 | Thermo Fisher Scientific | Hs02596875_s1 | |
| Sequence-based reagent | Taqman assay for MT-ND4 | Thermo Fisher Scientific | Hs02596876_g1 | |
| Sequence-based reagent | Taqman assay for MT-CO2 | Thermo Fisher Scientific | Hs02596865_g1 | |
| Sequence-based reagent | Taqman assay for MT-CO3 | Thermo Fisher Scientific | Hs02596866_g1 | |
| Sequence-based reagent | Taqman assay for HPRT-1 | Thermo Fisher Scientific | Hs99999909_m1 | |
| Sequence-based reagent | Taqman assay for TBP | Thermo Fisher Scientific | Hs00427620_m1 | |
| Sequence-based reagent | Taqman assay for RPL13A | Thermo Fisher Scientific | Hs03043885_g1 | |
| Commercial assay or kit | DNeasy Blood and Tissue kit | QIAGEN | 69506 | |
| Commercial assay or kit | Dual-Glo Stop&Glo reagents | Promega | E2920 | |

*Appendix 1 Continued on next page*

*Appendix 1 Continued*

| Reagent type (species) or resource | Designation | Source or reference | Identifiers | Additional information |
|---|---|---|---|---|
| Commercial assay or kit | Alt-R S.p. Cas9 D10A Nickase V3 | IDT | 1081058 | |
| Commercial assay or kit | Nucleofector reagent L | Lonza | VCA-1005 | |
| Commercial assay or kit | Alt-R HDR Enhancer V2 | IDT | 10007921 | |
| Commercial assay or kit | QuickExtract DNA Extraction Solution | Lucigen | QE09050 | |
| Commercial assay or kit | Alt-R Genome Editing Detection Kit | IDT | 1075932 | |
| Commercial assay or kit | Mitochondrial isolation kit | Thermo Fisher Scientific | 89874 | |
| Commercial assay or kit | NativePAGE Sample Prep Kit | Invitrogen | BN2008 | |
| Commercial assay or kit | Quant-iT PicoGreen dsDNA Assay Kit | Thermo Fisher Scientific | P7589 | |
| Commercial assay or kit | Triglyceride-Glo Assay kit | Promega | J3161 | |
| Commercial assay or kit | Glycerol-Glo Assay | Promega | J3150 | |
| Chemical compound, drug | Insulin | Sigma-Aldrich | I2643 | |
| Chemical compound, drug | Isoproterenol | Sigma-Aldrich | 1351005 | |
| Chemical compound, drug | Glucose | Angilent | 103577-100 | |
| Chemical compound, drug | Glutamine | Angilent | 103579-100 | |
| Chemical compound, drug | Carnitine | Sigma-Aldrich | C0283 | |
| Chemical compound, drug | Pyridine | Thermo Scientific | 019378.K2 | |
| Chemical compound, drug | *N*-Methyl-*N*-(trimethylsilyl) trifluoroacetamide | Thermo Scientific | A13141.22 | |
| Chemical compound, drug | Trimethylsilyl chloride | Thermo Scientific | A12535.30 | |
| Chemical compound, drug | 3-Nitrophenylhydrazine | Sigma-Aldrich | N21804 | |
| Chemical compound, drug | 1-Ethyl-3-(3-dimethylaminopropyl) carbodiimide hydrochloride | Thermo Scientific | 22980 | |
| Chemical compound, drug | Formic acid | Fisher Chemical | A117-50 | |
| Chemical compound, drug | Etomoxir | Sigma-Aldrich | E1905 | |
| Chemical compound, drug | Oligomycin | Sigma-Aldrich | O4876 | |
| Chemical compound, drug | FCCP | Sigma-Aldrich | C2920 | |
| Chemical compound, drug | Rotenone | Sigma-Aldrich | R8875 | |
| Chemical compound, drug | Antimycin A | Sigma-Aldrich | A8674 | |

