## [Editor Report]

In this study, Huang et al. perform detailed functional genomics assays in cultured adipocytes to provide mechanistic insight underlying an important obesity GWAS locus. These studies not only demonstrate allele-specific effects of MITF3 as a potential causal gene for variations in rs67785913 and rs1885988 alleles, but further provide a foundational framework from bridging GWAS associations to actionable pathways. The study strengths include genetic manipulation followed by detailed biochemical characterization to mimic and test the impacts of association, where future studies potentially involving in vivo characterizations could further inform the metabolic consequences of these observations.

---

## [Decision Letter]

**Decision letter after peer review:**

Thank you for submitting your article "Identification of a weight loss-associated causal eQTL in MTIF3 and the effects of MTIF3 deficiency on human adipocyte function" for consideration by *eLife*. Your article has been reviewed by 2 peer reviewers, and the evaluation has been overseen by a Reviewing Editor and David James as the Senior Editor. The following individual involved in the review of your submission has agreed to reveal their identity: Sean M. Hartig (Reviewer #1).

The reviewers have discussed their reviews with one another, and the Reviewing Editor has drafted this to help you prepare a revised submission. We all remain enthusiastic about the implications of these functional genomics experiments characterizing the locus and most requested revisions surround more detailed evaluation primarily in cell culture models used.

Essential revisions:

As you will see below, the primary suggestions for improving the study lie in more detailed molecular and functional characterization of MTIF3 and the effects of the locus. A few of these include protein quantification and staining (oil red O) for adipocytes where MTIF3 is perturbed. In addition, the reviewers requested a few text-based clarifications regarding the interpretation of the locus, listed below. Obviously, if you have any questions feel, please free to reach out.

*Reviewer #1 (Recommendations for the authors):*

It is important to point out that adipocyte differentiation and the expression of the associated metabolic programs critically contribute to the maintenance of insulin sensitivity in obesity. Defects in adipocyte differentiation derived from mitochondrial functions that reduce ATP synthesis and storage of surfeit carbons in fat tissue almost always associate with insulin resistance in animal models and people (see Trends Cell Biol 32:351-364, 2022; Cell. 185:419-446, 2022). The in vitro studies can add to understanding the important roles of MTIF3, but the authors should use more caution when extrapolating to in vivo effects of the eQTL.

*Reviewer #2 (Recommendations for the authors):*

1. The authors should give their hypothesis and interpretation of their results in the manuscript. There are no clear connections between results, which makes it difficult to understand the rationale of their experiments.

2. The labels of Figure 1 do not match the figure legends.

3. In figure 1F, what is the readout of the relative MTIF3expression? Are they based on western blot? If yes, it would be better to include the blog images.

4. In the method, the authors also added extra lipids in the differentiated medium to increase the lipid droplets. I'm curious about the effect of lipid deprivation in KO cells rather than glucose deprivation. The metabolomics showed the change of fatty acids at full medium. There might be some changes in fatty acid utilization in KO cells.

---

## [Author Response]

Essential revisions:As you will see below, the primary suggestions for improving the study lie in more detailed molecular and functional characterization of MTIF3 and the effects of the locus. A few of these include protein quantification and staining (oil red O) for adipocytes where MTIF3 is perturbed. In addition, the reviewers requested a few text-based clarifications regarding the interpretation of the locus, listed below. Obviously, if you have any questions feel, please free to reach out.

We thank for all the comments and suggestions the editors and reviewers made to help us improve our manuscript. We marked all the changes we made in the revised manuscript in yellow, and we also included an Appendix key resources table to list all the reagents we used in this study. Note that we also removed the p values and n (sample size) numbers in the main text, as they are already listed in the figures and figure legends. We also corrected some typos that were missed in the original manuscript. Furthermore, to meet the revision submission guidelines, we have included Key Resources Table as an appendix at the end of the manuscript, reformatted previous supplemental tables as Supplementary files 1a and 1b, and added legends to source files (uncropped blot images, and raw data files).

Reviewer #1 (Recommendations for the authors):It is important to point out that adipocyte differentiation and the expression of the associated metabolic programs critically contribute to the maintenance of insulin sensitivity in obesity. Defects in adipocyte differentiation derived from mitochondrial functions that reduce ATP synthesis and storage of surfeit carbons in fat tissue almost always associate with insulin resistance in animal models and people (see Trends Cell Biol 32:351-364, 2022; Cell. 185:419-446, 2022). The in vitro studies can add to understanding the important roles of MTIF3, but the authors should use more caution when extrapolating to in vivo effects of the eQTL.

We have now incorporated your suggestion, and expanded on it in the manuscript (lines 294-311).

Reviewer #2 (Recommendations for the authors):1. The authors should give their hypothesis and interpretation of their results in the manuscript. There are no clear connections between results, which makes it difficult to understand the rationale of their experiments.

Thank you for your suggestions. In the revised manuscript, we have included the hypotheses in the introduction (Lines: 70-76) and added or clarified the rationale behind the experiments in several paragraphs of the results (lines: 104-116, 138-140, 159-161, 170-171, 182-183, 199-200, 215-216). We also added more interpretation of our data to the discussion, among the highlighted parts, and the summary statement in lines 321-331.

2. The labels of Figure 1 do not match the figure legends.

We have corrected this error in the revised manuscript in the figure legend.

3. In figure 1F, what is the readout of the relative MTIF3expression? Are they based on western blot? If yes, it would be better to include the blog images.

In Figure 1F, the relative *MTIF3* expression represents mRNA level based on RT-qPCR assay. We have clarified that in the revised figure legend (lines 803, 806).

4. In the method, the authors also added extra lipids in the differentiated medium to increase the lipid droplets. I'm curious about the effect of lipid deprivation in KO cells rather than glucose deprivation. The metabolomics showed the change of fatty acids at full medium. There might be some changes in fatty acid utilization in KO cells.

We might not have been clear on this in the original manuscript – we only use extra lipids during the differentiation process. Then, after generating the knockout and during the restriction experiments, the adipocytes are maintained in medium *without* extra lipids, to ensure more efficient lipolysis. In other words, your suggestion for lipid deprivation has already been met in our experiments. We have clarified that in the methods (lines: 432-439) and in the results (line 192). As you suggest, the fatty acid utilization is affected by the *MTIF3* knockout, as demonstrated by the changed metabolite profile (Figure 7) and fatty acid oxidation (Figure 6).

References

1. Rudler, D.L., et al., *Fidelity of translation initiation is required for coordinated respiratory complex assembly.* Sci Adv, 2019. 5(12): p. eaay2118.

2. Renes, J., et al., *Calorie restriction-induced changes in the secretome of human adipocytes, comparison with resveratrol-induced secretome effects.* Biochim Biophys Acta, 2014. 1844(9): p. 1511-22.movie